# Simultaneous Confidence Bounds for Aggregated Effects via Exact Subset Optimization

Weihang Xu [* 1]   Huajie Qian [* 2]   Wotao Yin [2]   Xinshang Wang [2]

## Abstract

We study simultaneous confidence bounds for aggregated effects over downward-closed subset families of independent statistical tests. The bounds are obtained by bootstrap calibration of the maximum normalized aggregated effect over the relevant subset family, yielding valid post-hoc inference for data-selected subsets and tighter bounds than classical methods that protect all linear contrasts. A central challenge is that the required maximization is a nonlinear combinatorial optimization problem. We cast it as a weighted densest-subgraph problem and derive exact linear and mixed integer linear program reformulations, and we further develop a fully polynomial-time approximation scheme that exploits the rank-1 structure of the objective to scale to large families. On the statistical side, we establish a finite-sample coverage guarantee with a lighter dependence on the subset family size than high-dimensional central limit theorems provide. We illustrate the method on synthetic and real machine learning applications.

## 1. Introduction

Across many modern applications—large-scale A/B testing on non-overlapping subgroups (Kohavi et al., 2020; Bakshy et al., 2014), meta-analysis combining independent studies (Hedges & Olkin, 2014; Borenstein et al., 2009), performance evaluation across independent benchmark datasets (Dror et al., 2018; Arcuri & Briand, 2011; McGeoch, 2012; Demsar, 2006), and drug discovery (Iversen et al., 2012; Zhang et al., 1999)—analysts monitor collections of statistical tests whose per-test estimates are independent (or near-

independent) by construction. Each test yields an estimated effect with an estimated standard error. While individual tests can be weak or noisy, the aggregated effect over a subset of tests can be scientifically meaningful and more stable.

In practice, subsets are typically chosen from the data using procedures such as scan statistics, higher-criticism style screening, or gene-set enrichment (Donoho & Jin, 2004; Kulldorff, 1997; Walther, 2010; Arias-Castro et al., 2011; Subramanian et al., 2005). Because the reported subset is selected adaptively, naive inference for its aggregated effect is biased; we therefore need post-selection inference that remains valid for data-dependent choices of subsets.

Our goal is simultaneous inference for aggregated effects across downward-closed subset families—common examples include the family of all nonempty subsets and cardinality-constrained families. We construct confidence bounds that remain valid even when the reported subset is data-selected. This requires calibrating the distribution of the maximum aggregated statistic over the subset family, which is computationally challenging because the maximization is combinatorial and nonconvex.

We address the computational bottleneck by reducing the subset maximization to a weighted densest-subgraph problem and providing exact linear program (LP) and mixed integer linear program (MILP) reformulations. Exact evaluation of the maximum aggregated statistic is important: using a suboptimal value can cause undercoverage, while relying on loose dual bounds introduces unnecessary conservativeness. To scale beyond the regime where the exact constrained MILP remains tractable, we further develop a fully polynomial-time approximation scheme (FPTAS) that exploits the rank-1 structure of the subset-sum objective; a simple rescaling of the FPTAS output yields an upper bound on the maximum aggregated statistic with a controllable gap, which can still be used to produce tight and valid confidence bounds.

On the inferential side, we use bootstrap calibration of the subset-maximum statistic to obtain tight simultaneous bounds. Establishing finite-sample coverage is challenging here because the subset family can be exponentially large,

---

[*]Equal contribution  [1]Paul G. Allen School of Computer Science & Engineering, University of Washington, Seattle, US [2]Alibaba Group, Bellevue, US. Correspondence to: Huajie Qian <h.qian@alibaba-inc.com>.

*Proceedings of the 43rd International Conference on Machine Learning*, Seoul, South Korea. PMLR 306, 2026. Copyright 2026 by the author(s).

so standard high-dimensional Gaussian approximation arguments would only permit a less favorable growth of the family size. We instead bound the coverage error with a direct treatment exploiting the convexity of the relevant event set and the independence across tests. Moreover, our guarantee holds for finite bootstrap sizes that directly govern the computational cost, making the calibration practical.

Compared to projecting a confidence region for the full mean vector (which protects all linear contrasts) (Berk et al., 2013) or applying a Bonferroni correction across the subset family (which pays for its full cardinality and ignores its dependence structure) (Dunn, 1961), our approach of calibrating the maximum aggregated statistic is statistically more efficient because it targets only the subset-sum family that matters for our inferential goal and adapts to the underlying dependence structure; this typically yields substantially tighter bounds, as confirmed by our experiments.

**Contributions.** We (i) formalize simultaneous confidence bounds for aggregated effects over downward-closed subset families and the associated bootstrap calibration scheme that targets the subset-maximum statistic; (ii) give exact LP and MILP formulations that solve the required subset maximization for unconstrained and downward-closed linear families (including cardinality constraints), together with an FPTAS based on the rank-1 structure of the objective that scales to much larger problems while preserving the coverage guarantee; and (iii) establish a finite-sample coverage guarantee with favorable dependence on the subset family's log-cardinality and the bootstrap size, and demonstrate strong empirical performance.

## 2. Related Work

Aggregation-based detection has a long history in higher criticism and scan statistics (Donoho & Jin, 2004; Kulldorff, 1997; Walther, 2010; Arias-Castro et al., 2011), as well as gene-set enrichment and related pathway analyses (Subramanian et al., 2005). These methods primarily address detection, whereas we focus on simultaneous confidence bounds for data-selected subsets.

Post-selection inference offers several routes to valid inference after data-driven selection. Resampling-based multiple testing calibrates the tests jointly by resampling, yielding familywise error control (Westfall & Young, 1993; Romano & Wolf, 2005; Ge et al., 2003), but treats the hypotheses as a flat list, whereas our family is the set of all subset sums and statistically efficient calibration requires a combinatorial maximization. Selection-agnostic frameworks such as PoSI cover all contrasts induced by any selection step (Berk et al., 2013; Kuchibhotla, 2025), and conditional selective-inference methods give valid intervals given the selection event (Lee et al., 2016; Tibshirani et al., 2018); by targeting

only aggregated subset sums, we obtain substantially tighter bounds than protecting all linear contrasts. Sample splitting is also widely used (Cox, 1975; Wasserman & Roeder, 2009; Rasines & Young, 2023) but loses power and is infeasible when only per-test summary statistics are available (Kim et al., 2015; Guo & Wu, 2018).

Computationally, our subset maximization is related to two separate lines of combinatorial optimization. The densest-subgraph problem admits exact max-flow and LP algorithms (Goldberg, 1984; Charikar, 2000); our problem boils down to a more general setting where both edges and vertices are weighted, for which we establish analogous exact LP and MILP formulations. Separately, the 1D knapsack problem (with a linear objective) admits FPTAS (Ibarra & Kim, 1975; Chen et al., 2024); our objective is instead a nonlinear ratio, but its rank-1 structure can be utilized to design FPTAS in the same spirit as knapsack.

Lastly, coverage guarantees like ours are usually established by high-dimensional central limit theorems, which approximate the law of maxima of normalized sums over rectangles and sparsely convex sets (Chernozhukov et al., 2017; 2023). These results, however, are tailored to sample-mean statistics and, when applicable to our setting, only allow a slower growth rate in the subset family size relative to the per-test sample size compared to our direct treatment.

## 3. Simultaneous Inference for Aggregated Effects

We observe $m$ independent tests. Test $i \in \{1, \ldots, m\}$ targets an unknown scalar parameter $\theta_i$ using an estimator $\hat{\theta}_i$ with standard error estimate $\hat{\sigma}_i$. Let

$$\hat{\boldsymbol{\theta}} = (\hat{\theta}_1, \ldots, \hat{\theta}_m), \qquad \hat{\Sigma} = \text{diag}(\hat{\sigma}_1^2, \ldots, \hat{\sigma}_m^2). \quad (1)$$

For any nonempty subset $S \subseteq \{1, \ldots, m\}$, define the aggregated studentized statistic (the denominator sums per-test variances, valid under independence)

$$T(S; \hat{\boldsymbol{\theta}}, \hat{\Sigma}) = \frac{\sum_{i \in S} \hat{\theta}_i}{\sqrt{\sum_{i \in S} \hat{\sigma}_i^2}}.$$

The aggregated effect for a subset $S$ is the sum $\sum_{i \in S} \theta_i$. Throughout, we consider an admissible family of subsets that admits a downward-closed linear representation in the selectors $z_i = \mathbf{1}\{i \in S\}$: there exist $A, b$ such that $A$ has nonnegative entries and

$$\mathcal{A} = \left\{ S \subseteq \{1, \ldots, m\} : \boldsymbol{z} \in \{0,1\}^m, \ A\boldsymbol{z} \le b, \ \mathbf{1}^\top \boldsymbol{z} \ge 1 \right\},$$

which ensures downward closure (if $S \in \mathcal{A}$ then every nonempty $S' \subseteq S$ is also in $\mathcal{A}$). The cardinality constraint $1 \le |S| \le k$ is a special case with $A = \mathbf{1}^\top$ and $b = k$, together with the non-emptiness constraint. Our goal is a

$(1-\alpha)$-level lower bound for $\sum_{i \in S} \theta_i$ that is valid uniformly over data-dependent choices of $S \in \mathcal{A}$.

### 3.1. Bootstrap Calibration

Simultaneous inference is commonly obtained by calibrating the maximum of a test statistic over the relevant family. We estimate the null distribution of $\max_{S \in \mathcal{A}} T(S; \hat{\boldsymbol{\theta}}, \hat{\Sigma})$ via multiplier bootstrap using $N(0, \hat{\Sigma})$, and use its $(1-\alpha)$ quantile $c_{\mathrm{maxT}}$ to form a confidence bound in the form of

$$\sum_{i \in S} \hat{\theta}_i - c_{\mathrm{maxT}} \sqrt{\sum_{i \in S} \hat{\sigma}_i^2}$$

for any data-selected subset $S$. The full procedure is described in Algorithm 1.

---

**Algorithm 1** SubsetMaxT via Exact Optimization

---

**input** Variance estimates $\hat{\Sigma}$, admissible family $\mathcal{A}$ (linear constraints $Az \leq b$ and $\mathbf{1}^\top z \geq 1$), type-I error level $\alpha$, bootstrap size $B$.

1: **for** $b = 1, \dots, B$ **do**
2:     Sample $\hat{\boldsymbol{\theta}}^b \sim N(0, \hat{\Sigma})$.
3:     Compute $T_b^{\max} = \max(0, \max_{S \in \mathcal{A}} T(S; \hat{\boldsymbol{\theta}}^b, \hat{\Sigma}))$ using the exact optimizer in Section 4.
4: **end for**
5: Output $c_{\mathrm{maxT}}$ to be the $\lceil (1 - \alpha)(B + 1) \rceil$-th order statistic of $\{T_b^{\max}\}_{b=1}^B$.

---

## 4. Subset Optimization

The bootstrap calibration in SubsetMaxT (Algorithm 1) requires solving

$$\max_{S \in \mathcal{A}} T(S; \hat{\boldsymbol{\theta}}, \hat{\Sigma}),$$

which is a nonlinear combinatorial optimization. This section focuses on efficient algorithms for solving this optimization. To proceed, introduce binary selectors $z_i \in \{0, 1\}$ with $z_i = 1 \Leftrightarrow i \in S$, we can write

$$
\begin{aligned}
\max \quad & \frac{\sum_{i=1}^m \hat{\theta}_i z_i}{\sqrt{\sum_{i=1}^m \hat{\sigma}_i^2 z_i}} \\
\text{s.t.} \quad & Az \leq b, \quad \mathbf{1}^\top z \geq 1, \quad z_i \in \{0, 1\}.
\end{aligned}
\tag{2}
$$

### 4.1. Reduction to a Densest Subgraph Problem

We first describe a key monotonicity property that allows us to restrict to $\hat{\theta}_i$'s of the same sign when solving (2).

**Proposition 4.1** (Sign screening). *Assume $\hat{\sigma}_i > 0$ for all $i$. Let $S \neq \varnothing$ and $i \notin S$. If $T(S; \hat{\theta}, \hat{\Sigma}) > 0$ and $\hat{\theta}_i \leq 0$, then $T(S \cup \{i\}; \hat{\theta}, \hat{\Sigma}) < T(S; \hat{\theta}, \hat{\Sigma})$. Conversely, if $T(S; \hat{\theta}, \hat{\Sigma}) < 0$ and $\hat{\theta}_i \geq 0$, then $T(S \cup \{i\}; \hat{\theta}, \hat{\Sigma}) > T(S; \hat{\theta}, \hat{\Sigma})$.*

The proof is given in Appendix A. The intuition is that when $T(S; \hat{\theta}, \hat{\Sigma}) > 0$ the numerator $\sum_{j \in S} \hat{\theta}_j$ is positive, so adding a test with $\hat{\theta}_i \leq 0$ does not increase the numerator yet strictly enlarges the denominator $\sqrt{\sum_{j \in S} \hat{\sigma}_j^2}$, strictly lowering $T$. A symmetric argument handles the case $T(S; \hat{\theta}, \hat{\Sigma}) < 0$ with $\hat{\theta}_i \geq 0$.

By Proposition 4.1, the optimal subset $S^*$ for the maximization problem in (2) must satisfy $i \notin S^*$ whenever $\hat{\theta}_i \leq 0$, and hence it suffices to consider tests with $\hat{\theta}_i > 0$ only. Note that in the case where $\hat{\theta}_i \leq 0$ for all $i$, the maximizing subset is simply the maximizing singleton as the following proposition shows:

**Proposition 4.2.** *If $\hat{\theta}_i \leq 0$ for all $i$, let $i^* \in \arg\max_{1 \leq i \leq m \text{ s.t. } A_i \leq b} \hat{\theta}_i / \hat{\sigma}_i$, where $A_i$ is the $i$-th column of $A$, then $z_{i^*} = 1, z_i = 0, \forall i \neq i^*$ is an optimal solution to the maximization in (2).*

The proof is given in Appendix B. Intuitively, when all $\hat{\theta}_i \leq 0$ the best single-test ratio $\hat{\theta}_{i^*} / \hat{\sigma}_{i^*}$ is itself nonpositive and pooling tests can only decrease the statistic.

Without loss of generality, we can now assume $\hat{\theta}_i > 0$ for all $i$. However, problem (2) is still a mixed integer quadratic programming problem, thus NP-complete in general. To efficiently solve this challenging problem, we introduce a novel graph-theory-based optimization approach.

Squaring the objective (which preserves the maximizer when the objective is non-negative) yields

$$
\max_{z \in \{0,1\}^m} \frac{\sum_{i=1}^m \sum_{j=1}^m \hat{\theta}_i \hat{\theta}_j \, z_i z_j}{\sum_{i=1}^m \hat{\sigma}_i^2 z_i} \quad \text{s.t.} \quad Az \leq b, \mathbf{1}^\top z \geq 1.
\tag{3}
$$

Our first key observation is that (3) could be seen as a constrained weighted densest-subgraph problem. Consider a complete undirected graph $G = (V, E)$ with $V = \{1, \dots, m\}$, edge weights $w_{ij} = \hat{\theta}_i \hat{\theta}_j$ for the edge $(i, j)$ (allowing loops $\hat{\theta}_i^2$ for edge $(i, i)$), and vertex weights $c_i = \hat{\sigma}_i^2$. The objective can be interpreted as the density $\sum_{i,j \in S} w_{ij} / \sum_{i \in S} c_i$ of the subgraph $S = \{i : z_i = 1\}$, i.e., the ratio of its total edge weights to its total vertex weights.

### 4.2. Unconstrained Case

We first consider the unconstrained subset family. In this case, the unweighted densest subgraph problem admits polynomial-time algorithms based on network flow (Goldberg, 1984) or LP reformulation (Charikar, 2000). However, previous results (*e.g.*, (Goldberg, 1984; Fazzone et al., 2022)) cannot be directly applied to solve the general weighted optimization (3). Our second key observation here is that the LP-based algorithm for the unweighted problem can be generalized to solve the weighted version (3):

**Theorem 4.3** (LP equivalence and tractability). *Assume $\hat{\theta}_i > 0$ and $\hat{\sigma}_i^2 > 0$ for all $i$ and $A, b$ are empty. The linear program (LP)*

$$\max \quad \sum_{i=1}^m \sum_{j=1}^m \hat{\theta}_i \hat{\theta}_j \, v_{ij}$$

$$\begin{aligned} s.t. \quad & v_{ij} \leq u_i, \quad \forall\, i, j, \\ & v_{ij} \leq u_j, \quad \forall\, i, j, \\ & \sum_{i=1}^m \hat{\sigma}_i^2 \, u_i = 1, \\ & u_i \geq 0, \; v_{ij} \geq 0, \quad \forall\, i, j. \end{aligned} \qquad (4)$$

*with decision variables $u_i, v_{i,j}$ for $i, j \in \{1, 2, \ldots, m\}$ has the same optimal value as (3). Moreover, for any optimal solution $u_i^*, v_{ij}^*$ of the LP, there exists some $i^* \in \{1, 2, \ldots, m\}$ such that $z_i := \mathbf{1}(u_i^* \geq u_{i^*}^*), i \in \{1, 2, \ldots, m\}$ is an optimal solution to (3).*

Formulation (4) is the weighted analogue of the classic LP for densest subgraph, adapted to the edge-over-vertex ratio induced by the pooled $t$-statistic. The equivalence in Theorem 4.3 has two essential ingredients. First, a variable substitution $u_i = z_i / \sum_j \hat{\sigma}_j^2 z_j$ and $v_{ij} = \min(u_i, u_j)$ (which equals $z_i z_j / \sum_j \hat{\sigma}_j^2 z_j$ when $z_i \in \{0, 1\}$) maps every binary $\boldsymbol{z}$ to a feasible LP solution with matching objective, so the LP value is at least the discrete optimum. Second, the LP relaxation does not inflate the optimum: for any LP optimum $u^*$, a thresholded subset $\{i : u_i^* \geq r\}$ for a certain $r$ can be shown to attain at least the LP value in the objective through a parametric averaging argument analogous to that in Charikar (2000) for the unweighted case. To solve unconstrained (3), one can solve (4) with off-the-shelf LP solvers and convert the optimal solution back to an optimum of (3) by Theorem 4.3. Proof of Theorem 4.3 can be found in Appendix C.

### 4.3. Constrained Case

In the presence of downward-closed linear constraints $A\boldsymbol{z} \leq b$ with nonnegative $A$, the LP formulation no longer works, but these constraints can be enforced by augmenting the LP with binary support variables. An exact MILP is

$$\max \quad \sum_{i=1}^m \sum_{j=1}^m \hat{\theta}_i \hat{\theta}_j v_{ij}$$

$$\begin{aligned} s.t. \quad & v_{ij} \leq u_i, \quad v_{ij} \leq u_j, \quad \forall i, j, \\ & \sum_{i=1}^m \hat{\sigma}_i^2 u_i = 1, \\ & 0 \leq \hat{\sigma}_i^2 u_i \leq z_i, \quad \forall i, \\ & A\boldsymbol{z} \leq b, \\ & z_i \in \{0, 1\}, \; v_{ij} \geq 0. \end{aligned} \qquad (5)$$

The following result shows that the MILP is exact for downward-closed subset families.

**Theorem 4.4** (MILP exactness for downward-closed families). *Assume $\hat{\theta}_i > 0$ and $\hat{\sigma}_i^2 > 0$ for all $i$, and let $A \geq 0$ element-wise. The MILP above has the same optimal value as (3). Moreover, for any optimal MILP solution $(\boldsymbol{u}^*, \boldsymbol{v}^*, \boldsymbol{z}^*)$, there exists an index $i^*$ such that the subset defined by $z_i := \mathbf{1}\{u_i^* \geq u_{i^*}^*, z_i^* = 1\}$ is feasible and optimal for (3).*

When $A\boldsymbol{z} \leq b$ encodes cardinality constraints, this reduces to the cardinality-constrained MILP; when $A\boldsymbol{z} \leq b$ is absent (all nonempty subsets), relaxing $z_i \in \{0, 1\}$ recovers the LP in Theorem 4.3. For downward-closed linear families, the MILP can be solved by standard branch-and-bound solvers. Proof of Theorem 4.4 is in Appendix D.

### 4.4. FPTAS for Constrained Families

For large-scale problems, solving the MILP can be expensive (see runtime measurements in Section 6.3). However, the rank-1 structure of the edge weights $w_{ij} = \hat{\theta}_i \hat{\theta}_j$ implies that the squared objective $T(S; \hat{\theta}, \hat{\Sigma})^2 = P(S)^2/Q(S)$ depends on the selectors only through the two scalar aggregates $P(S) = \sum_{i \in S} \hat{\theta}_i$ and $Q(S) = \sum_{i \in S} \hat{\sigma}_i^2$. This 2D sufficient statistic enables a dynamic-programming FPTAS that discretizes one aggregate and tracks the other exactly, structurally analogous to the classical knapsack FPTAS (Ibarra & Kim, 1975) but with a nonlinear ratio objective.

**Theorem 4.5** (FPTAS for the cardinality-constrained case). *Fix $\epsilon \in (0, 1)$. For the cardinality-constrained problem $\max_{|S| \leq k} T(S; \hat{\boldsymbol{\theta}}, \hat{\Sigma})$ with $\hat{\theta}_i, \hat{\sigma}_i^2 > 0$, there is an algorithm that returns a value $\widetilde{T}$ satisfying $(1 - \epsilon) T^* \leq \widetilde{T} \leq T^*$ in time $O(mk^3/\epsilon)$ (up to a logarithmic factor in the input encoding length and the cardinality bound $k$), where $T^*$ is the true maximum.*

Because the FPTAS underestimates $T^*$, plugging $\widetilde{T}$ directly into the bootstrap calibration of Algorithm 1 would underestimate the critical value and risk undercoverage. Rescaling each bootstrap statistic by $1/(1 - \epsilon)$, i.e., using $\widehat{T} := \widetilde{T}/(1 - \epsilon) \geq T^*$, restores a valid lower bound at the cost of at most an $O(\epsilon)$-multiplicative inflation of the critical value. Full algorithm and proof, together with a discussion of the case with a single downward-closed weight constraint, are in Appendix G.

## 5. Finite-Sample Coverage Guarantees

We present a finite-sample coverage theorem.

**Theorem 5.1** (Finite-Sample Coverage Error). *Denote by $\sigma_i^2 := \mathrm{Var}(\hat{\theta}_i)$ for all $i$. Assume $\sigma_i^2 \in (0, \infty)$ and $\hat{\sigma}_i^2 \in (0, \infty)$ almost surely for all $i$. For every $\boldsymbol{\theta} \in \mathbb{R}^m$, $\alpha \in (0, \frac{1}{2})$ and $B \geq \frac{1-\alpha}{\alpha}$, the critical value $c_{\mathrm{maxT}}$ from Algorithm 1*

*satisfies*

$$\mathbb{P}\left(\max_{S\in\mathcal{A}} T(S;\hat{\boldsymbol{\theta}}-\boldsymbol{\theta},\hat{\Sigma}) \le c_{\mathrm{maxT}}\right) \ge 1-\alpha-\mathcal{E}$$

*with*

$$\mathcal{E} = \sum_{i=1}^{m}\sup_{x\in\mathbb{R}}\left|\mathbb{P}\left(\frac{\hat{\theta}_i-\theta_i}{\hat{\sigma}_i}\le x\right)-\Phi(x)\right|$$
$$+\inf_{\delta\in(0,1)}\left[\mathbb{P}\left(\max_i\left|\frac{\hat{\sigma}_i^2}{\sigma_i^2}-1\right|\ge\delta\right)+\right.$$
$$\left. C\delta\left(\log(|\mathcal{A}|/\alpha)+e^{-B\alpha^2/2}\sqrt{\log(B)\log(|\mathcal{A}|)}\right)\right],$$

*where $C$ is a universal constant, $|\mathcal{A}|$ denotes the cardinality of $\mathcal{A}$, and $\Phi(\cdot)$ is the CDF of the standard normal.*

Theorem 5.1 lower bounds the finite-sample coverage of the SubsetMaxT by $1-\alpha$ up to an explicit error term $\mathcal{E}$. The first component of $\mathcal{E}$, $\sum_{i=1}^{m}\sup_x\left|\Pr((\hat{\theta}_i-\theta_i)/\hat{\sigma}_i\le x)-\Phi(x)\right|$, captures the marginal non-Gaussianity of each studentized coordinate; it vanishes if each $(\hat{\theta}_i-\theta_i)/\hat{\sigma}_i$ is exactly standard normal. The second component controls the impact of using estimated variances in both the statistic and the bootstrap calibration. Notably, this bound holds for finite bootstrap size $B$, with the dependence on $B$ entering only through the rapidly decaying term $e^{-B\alpha^2/2}\sqrt{\log(B)\log(|\mathcal{A}|)}$. This matters in practice because $B$ directly determines the computational cost of the procedure, and the exponential decay in $B$ suggests a moderate or even small $B$ already suffices. Moreover, the $B$ per-replicate maximizations are independent and hence can be easily parallelized to accelerate computation. We obtain it by bounding the gap between the empirical and population $(1-\alpha)$-quantiles of the resampled statistics via a Dvoretzky–Kiefer–Wolfowitz inequality (Appendix E, Lemma E.6). The proof of Theorem 5.1 bounds the error by a direct examination of the probabilities, exploiting the convexity of the event set and the independence among tests. The detailed proof is left to Appendix E.

In the canonical case where each $\hat{\theta}_i$ is the studentized statistic computed from i.i.d. data, the bound from Theorem 5.1 can be further simplified:

**Corollary 5.2.** *Assume that for each $i$, $\hat{\theta}_i$ is the sample mean of $n_i$ i.i.d. observations with mean $\theta_i$, and $\hat{\sigma}_i^2$ is the usual standard-error estimate. Suppose the following two rate conditions hold for some constants $C_{\mathrm{BE}}, c > 0$:*

$$\sup_{x\in\mathbb{R}}\left|\mathbb{P}\left(\frac{\hat{\theta}_i-\theta_i}{\hat{\sigma}_i}\le x\right)-\Phi(x)\right|\le\frac{C_{\mathrm{BE}}}{\sqrt{n_i}},$$
$$\mathbb{P}\left(\left|\frac{\hat{\sigma}_i^2}{\sigma_i^2}-1\right|\ge\delta\right)\le 2\exp(-c\,n_i\delta^2),\qquad\forall\delta\in(0,1).$$

*Then, with $n_{\min} := \min_i n_i$ and $L := \log(|\mathcal{A}|/\alpha) + e^{-B\alpha^2/2}\sqrt{\log(B)\log(|\mathcal{A}|)}$, the error term $\mathcal{E}$ in Theorem 5.1 satisfies*

$$\mathcal{E} \le C_{\mathrm{BE}}\sum_{i=1}^{m}\frac{1}{\sqrt{n_i}}+\inf_{\delta\in(0,1)}\left[2m\exp(-c\,n_{\min}\delta^2)+C\,\delta\,L\right].$$

*In particular, whenever $n_{\min} > \log(mn_{\min})/c$, choosing $\delta = \sqrt{\frac{\log(mn_{\min})}{c\,n_{\min}}} < 1$ yields the explicit bound*

$$\mathcal{E} \le C_{\mathrm{BE}}\sum_{i=1}^{m}\frac{1}{\sqrt{n_i}} + \frac{2}{n_{\min}} + C\,L\sqrt{\frac{\log(mn_{\min})}{c\,n_{\min}}}.$$

The proof is in Appendix F. Corollary 5.2 shows that the coverage error splits into a per-test Gaussian-approximation cost $\sum_i 1/\sqrt{n_i}$, present even for a single fixed subset, and a calibration error of order $\log|\mathcal{A}|/\sqrt{n_{\min}}$ (up to logarithmic factors) that is the only place the subset family enters. The calibration is therefore accurate once $\log|\mathcal{A}|$ grows slower than $\sqrt{n_{\min}}$. For the unconstrained family $\log|\mathcal{A}|\asymp m$, so this amounts to $m = o(\sqrt{n_{\min}})$, but it is far weaker for cardinality-constrained families, where $\log|\mathcal{A}| = O(k\log m)$. This regime is natural in our target applications (e.g., subgroup A/B tests with large per-subgroup samples, ML benchmark comparisons with a moderate number of datasets); the condition is only sufficient, however, and empirically the method continues to deliver valid coverage even when $\log|\mathcal{A}|$ is comparable to or exceeds $\sqrt{n_{\min}}$.

A natural alternative is to approximate the law of $\max_{S\in\mathcal{A}} T(S;\hat{\theta},\hat{\Sigma})$ by a high-dimensional CLT for maxima (Chernozhukov et al., 2017; 2023). This has two drawbacks in our setting. First, such results approximate sums of independent vectors and therefore require each $\hat{\theta}_i$ to be a sample mean (or asymptotically linear), whereas Theorem 5.1 only uses independence across tests and covers arbitrary per-test statistics. Second, even when each $\hat{\theta}_i$ is a sample mean, viewing the problem as the maximum of $|\mathcal{A}|$ linear forms yields a coverage error of order $\log^{7/6}|\mathcal{A}|\,n_{\min}^{-1/6}$ classically (Chernozhukov et al., 2017), or $\log^{3/2}|\mathcal{A}|\,n_{\min}^{-1/2}$ up to logarithmic factors under the sharpest nearly-optimal variants (Chernozhukov et al., 2023). These permit the family to grow only as fast as $\log|\mathcal{A}| = o(n_{\min}^{1/7})$ and $\log|\mathcal{A}| = o(n_{\min}^{1/3})$ respectively, whereas our bound permits the strictly faster $\log|\mathcal{A}| = o(\sqrt{n_{\min}})$.

## 6. Experiments

We evaluate the tightness and validity of the proposed bounds on synthetic data and on machine-learning applications where aggregated effects over structured subsets are scientifically meaningful.

## 6.1. Experiment Design

We use the same subset search algorithm $\mathcal{S}$ for all methods to select a data-dependent subset, and then compute and compare confidence bounds for the aggregated effect on the same subset with different methods using the same data. We compare our bounds based on exact subset optimization (SubsetMaxT) and FPTAS approximation (SubsetMaxT_FPTAS, $\epsilon = 0.01$) with the Scheffé-type bound and the Bonferroni bound. When our SubsetMaxT method is noted with "unconstr." we mean applying SubsetMaxT as if the subset family $\mathcal{A}$ contains all non-empty subsets. For the subset search algorithm, we use a greedy subset scanning based on Gaussian likelihood ratio, a common choice in the scan-statistics literature (Neill, 2012; Kulldorff, 1997). A detailed description of the two baseline methods and the subset search algorithm can be found in Appendix H.1. Appendix H.3 compares alternative search algorithms and confirms that coverage is invariant to this choice, consistent with Theorem 5.1.

For each test scenario, we perform many independent runs for each method to obtain many replicates of lower confidence bounds with a nominal level 95%, and then report empirical coverage and the mean of the bounds with standard errors. We set bootstrap size $B$ and replicate count uniformly across methods.

## 6.2. Synthetic Experiments

We simulate $m$ independent tests with per-test samples $\{X_{i\ell}\}_{\ell=1}^{50}$, set $\hat{\theta}_i$ to the sample mean and $\hat{\sigma}_i^2$ to the sample variance divided by 50, and vary the number of tests $m$ and the cardinality limit $k$ to explore small and moderate regimes. We consider the following patterns for the ground truth:

- **Signal pattern:** We consider both sparse (only 10% tests having $\theta_i > 0$) and dense (80% tests having $\theta_i > 0$).
- **Noise pattern:** We consider both homogeneous and heterogeneous noise levels across tests.

resulting in a total of 4 signal-noise pattern combinations.

**Synthetic results summary.** Tables 1–4 compare SubsetMaxT to Scheffé and Bonferroni across sparse/dense signal and homogeneous/heterogeneous lognormal noise settings, for multiple $(m, k)$ pairs. Across these configurations, all three methods achieve empirical coverage at (and in these runs often above) the nominal 95% level, so the primary difference is tightness: SubsetMaxT typically yields the largest (least conservative) lower bounds, Scheffé is intermediate, and Bonferroni is the most conservative. The gains of SubsetMaxT over the baselines are most pronounced for moderate-to-large $k$ and in dense-signal regimes, where it can improve the bound mean substantially

*Table 1.* Synthetic results under sparse signals and homogeneous lognormal noises. Bootstrap size $B = 50$, #replicates = 200. SubsetMaxT_FPTAS uses $\epsilon = 0.01$ (see Section 4.4 and Appendix G).

| m | k | Method | Coverage | Bound Mean | Bound SE |
|---|---|---|---|---|---|
| 30 | 5 | SubsetMaxT | 1.000 | 1.37e+00 | 1.41e-02 |
| | | SubsetMaxT_FPTAS | 1.000 | 1.36e+00 | 1.41e-02 |
| | | SubsetMaxT(unconstr.) | 1.000 | 1.29e+00 | 1.39e-02 |
| | | Scheffé | 1.000 | 1.31e+00 | 1.40e-02 |
| | | Bonferroni | 1.000 | 1.24e+00 | 1.38e-02 |
| 30 | 10 | SubsetMaxT | 1.000 | 1.30e+00 | 1.35e-02 |
| | | SubsetMaxT_FPTAS | 1.000 | 1.29e+00 | 1.35e-02 |
| | | SubsetMaxT(unconstr.) | 1.000 | 1.29e+00 | 1.35e-02 |
| | | Scheffé | 1.000 | 1.20e+00 | 1.38e-02 |
| | | Bonferroni | 1.000 | 1.01e+00 | 1.43e-02 |
| 30 | 30 | SubsetMaxT | 1.000 | 1.29e+00 | 1.35e-02 |
| | | Scheffé | 1.000 | 1.17e+00 | 1.40e-02 |
| | | Bonferroni | 1.000 | 8.99e-01 | 1.50e-02 |
| 50 | 5 | SubsetMaxT | 1.000 | 2.25e+00 | 1.75e-02 |
| | | SubsetMaxT_FPTAS | 1.000 | 2.23e+00 | 1.75e-02 |
| | | SubsetMaxT(unconstr.) | 1.000 | 1.98e+00 | 1.74e-02 |
| | | Scheffé | 1.000 | 2.18e+00 | 1.78e-02 |
| | | Bonferroni | 1.000 | 2.06e+00 | 1.68e-02 |
| 50 | 10 | SubsetMaxT | 1.000 | 2.09e+00 | 1.69e-02 |
| | | SubsetMaxT_FPTAS | 1.000 | 2.07e+00 | 1.69e-02 |
| | | SubsetMaxT(unconstr.) | 1.000 | 2.01e+00 | 1.68e-02 |
| | | Scheffé | 1.000 | 1.97e+00 | 1.77e-02 |
| | | Bonferroni | 1.000 | 1.66e+00 | 1.76e-02 |
| 50 | 50 | SubsetMaxT | 1.000 | 2.02e+00 | 1.69e-02 |
| | | Scheffé | 1.000 | 1.80e+00 | 1.78e-02 |
| | | Bonferroni | 1.000 | 1.20e+00 | 2.07e-02 |
| 100 | 5 | SubsetMaxT_FPTAS | 1.000 | 4.30e+00 | 3.79e-02 |
| | | SubsetMaxT(unconstr.) | 1.000 | 3.77e+00 | 3.79e-02 |
| | | Scheffé | 1.000 | 4.26e+00 | 3.91e-02 |
| | | Bonferroni | 1.000 | 4.16e+00 | 3.85e-02 |
| 100 | 10 | SubsetMaxT_FPTAS | 1.000 | 7.04e+00 | 2.86e-02 |
| | | SubsetMaxT(unconstr.) | 1.000 | 6.61e+00 | 2.85e-02 |
| | | Scheffé | 1.000 | 6.93e+00 | 3.02e-02 |
| | | Bonferroni | 1.000 | 6.50e+00 | 2.74e-02 |
| 100 | 100 | SubsetMaxT | 1.000 | 6.62e+00 | 2.86e-02 |
| | | Scheffé | 1.000 | 6.22e+00 | 2.83e-02 |
| | | Bonferroni | 1.000 | 4.84e+00 | 2.72e-02 |

(sometimes changing the bound from negative to positive), while retaining the same coverage behavior. When $m$ is large and $k$ is small, SubsetMaxT(unconstr.) can underperform Scheffé/Bonferroni, since it must protect all subsets (effectively paying for a much larger multiplicity), whereas the cardinality-constrained SubsetMaxT continues to outperform by tailoring the correction to the restricted family. All four tables also report SubsetMaxT_FPTAS. At $m = 100$ the exact cardinality-constrained MILP becomes too slow to compute, so only SubsetMaxT_FPTAS is reported there; it remains the tightest method, beating Scheffé and Bonferroni in bound mean. Appendix H.4 extends this comparison to $m$ up to 5000, where SubsetMaxT_FPTAS remains the tightest method. Standard errors are comparable across methods within each $(m, k)$ block, suggesting the observed gaps in bound means reflect systematic tightness differences rather than Monte Carlo noise.

Additional results for normal noises can be found in Tables 10–13 in the appendix.

*Table 2.* Synthetic results under sparse signals and heterogeneous lognormal noises. Bootstrap size $B = 50$, #replicates $= 200$.

| m | k | Method | Coverage | Bound Mean | Bound SE |
|---|---|--------|----------|------------|----------|
| | | SubsetMaxT | 1.000 | 2.22e+00 | 1.24e-02 |
| | | SubsetMaxT_FPTAS | 1.000 | 2.21e+00 | 1.23e-02 |
| 30 | 5 | SubsetMaxT(unconstr.) | 1.000 | 2.16e+00 | 1.21e-02 |
| | | Scheffé | 1.000 | 2.15e+00 | 1.17e-02 |
| | | Bonferroni | 1.000 | 2.10e+00 | 1.13e-02 |
| | | SubsetMaxT | 1.000 | 2.16e+00 | 1.21e-02 |
| | | SubsetMaxT_FPTAS | 1.000 | 2.16e+00 | 1.20e-02 |
| 30 | 10 | SubsetMaxT(unconstr.) | 1.000 | 2.16e+00 | 1.21e-02 |
| | | Scheffé | 1.000 | 2.06e+00 | 1.15e-02 |
| | | Bonferroni | 1.000 | 1.92e+00 | 1.20e-02 |
| | | SubsetMaxT | 1.000 | 2.16e+00 | 1.21e-02 |
| 30 | 30 | Scheffé | 1.000 | 2.04e+00 | 1.16e-02 |
| | | Bonferroni | 1.000 | 1.83e+00 | 1.31e-02 |
| | | SubsetMaxT | 1.000 | 1.83e+00 | 2.34e-02 |
| | | SubsetMaxT_FPTAS | 1.000 | 1.82e+00 | 2.33e-02 |
| 50 | 5 | SubsetMaxT(unconstr.) | 1.000 | 1.61e+00 | 2.24e-02 |
| | | Scheffé | 1.000 | 1.73e+00 | 2.29e-02 |
| | | Bonferroni | 1.000 | 1.62e+00 | 2.17e-02 |
| | | SubsetMaxT | 1.000 | 1.69e+00 | 2.24e-02 |
| | | SubsetMaxT_FPTAS | 1.000 | 1.67e+00 | 2.23e-02 |
| 50 | 10 | SubsetMaxT(unconstr.) | 1.000 | 1.62e+00 | 2.21e-02 |
| | | Scheffé | 1.000 | 1.52e+00 | 2.15e-02 |
| | | Bonferroni | 1.000 | 1.25e+00 | 2.01e-02 |
| | | SubsetMaxT | 1.000 | 1.62e+00 | 2.21e-02 |
| 50 | 50 | Scheffé | 1.000 | 1.37e+00 | 2.11e-02 |
| | | Bonferroni | 1.000 | 8.34e-01 | 2.11e-02 |
| | | SubsetMaxT_FPTAS | 1.000 | 3.02e+00 | 2.21e-02 |
| 100 | 5 | SubsetMaxT(unconstr.) | 1.000 | 2.59e+00 | 2.07e-02 |
| | | Scheffé | 1.000 | 2.95e+00 | 2.22e-02 |
| | | Bonferroni | 1.000 | 2.86e+00 | 2.18e-02 |
| | | SubsetMaxT_FPTAS | 1.000 | 5.62e+00 | 5.15e-02 |
| 100 | 10 | SubsetMaxT(unconstr.) | 1.000 | 5.23e+00 | 4.83e-02 |
| | | Scheffé | 1.000 | 5.40e+00 | 5.00e-02 |
| | | Bonferroni | 1.000 | 4.98e+00 | 4.58e-02 |
| | | SubsetMaxT | 1.000 | 5.24e+00 | 4.82e-02 |
| 100 | 100 | Scheffé | 1.000 | 4.71e+00 | 4.38e-02 |
| | | Bonferroni | 1.000 | 3.34e+00 | 3.52e-02 |

*Table 3.* Synthetic results under dense signals and homogeneous lognormal noises. Bootstrap size $B = 50$, #replicates $= 200$.

| m | k | Method | Coverage | Bound Mean | Bound SE |
|---|---|--------|----------|------------|----------|
| | | SubsetMaxT | 1.000 | -1.90e-01 | 1.22e-02 |
| | | SubsetMaxT_FPTAS | 1.000 | -2.03e-01 | 1.22e-02 |
| 30 | 5 | SubsetMaxT(unconstr.) | 1.000 | -3.03e-01 | 1.21e-02 |
| | | Scheffé | 1.000 | -2.69e-01 | 1.21e-02 |
| | | Bonferroni | 1.000 | -3.67e-01 | 1.05e-02 |
| | | SubsetMaxT | 1.000 | -9.37e-02 | 2.08e-02 |
| | | SubsetMaxT_FPTAS | 1.000 | -1.14e-01 | 2.08e-02 |
| 30 | 10 | SubsetMaxT(unconstr.) | 1.000 | -1.08e-01 | 2.08e-02 |
| | | Scheffé | 1.000 | -2.91e-01 | 2.10e-02 |
| | | Bonferroni | 1.000 | -6.68e-01 | 1.85e-02 |
| | | SubsetMaxT | 1.000 | -9.53e-03 | 2.71e-02 |
| 30 | 30 | Scheffé | 1.000 | -3.11e-01 | 2.67e-02 |
| | | Bonferroni | 1.000 | -9.72e-01 | 2.49e-02 |
| | | SubsetMaxT | 1.000 | -2.15e-01 | 1.16e-02 |
| | | SubsetMaxT_FPTAS | 1.000 | -2.28e-01 | 1.16e-02 |
| 50 | 5 | SubsetMaxT(unconstr.) | 1.000 | -4.80e-01 | 1.18e-02 |
| | | Scheffé | 1.000 | -2.83e-01 | 1.17e-02 |
| | | Bonferroni | 1.000 | -3.97e-01 | 1.03e-02 |
| | | SubsetMaxT | 1.000 | -1.28e-01 | 1.89e-02 |
| | | SubsetMaxT_FPTAS | 1.000 | -1.51e-01 | 1.89e-02 |
| 50 | 10 | SubsetMaxT(unconstr.) | 1.000 | -2.34e-01 | 1.87e-02 |
| | | Scheffé | 1.000 | -2.97e-01 | 1.85e-02 |
| | | Bonferroni | 1.000 | -7.21e-01 | 1.65e-02 |
| | | SubsetMaxT | 1.000 | 3.10e-01 | 4.10e-02 |
| 50 | 50 | Scheffé | 1.000 | -1.91e-01 | 3.96e-02 |
| | | Bonferroni | 1.000 | -1.62e+00 | 3.93e-02 |
| | | SubsetMaxT_FPTAS | 1.000 | -2.59e-01 | 1.02e-02 |
| 100 | 5 | SubsetMaxT(unconstr.) | 1.000 | -8.68e-01 | 9.49e-03 |
| | | Scheffé | 1.000 | -3.06e-01 | 1.04e-02 |
| | | Bonferroni | 1.000 | -4.21e-01 | 9.13e-03 |
| | | SubsetMaxT_FPTAS | 1.000 | -2.53e-01 | 1.60e-02 |
| 100 | 10 | SubsetMaxT(unconstr.) | 1.000 | -6.95e-01 | 1.48e-02 |
| | | Scheffé | 1.000 | -3.65e-01 | 1.61e-02 |
| | | Bonferroni | 1.000 | -8.03e-01 | 1.37e-02 |
| | | SubsetMaxT | 1.000 | 1.05e+00 | 5.17e-02 |
| 100 | 100 | Scheffé | 1.000 | 8.60e-02 | 4.99e-02 |
| | | Bonferroni | 1.000 | -3.23e+00 | 4.54e-02 |

## 6.3. Robustness, Scalability, and Bootstrap Size

We summarize three practical studies whose full tables are deferred to Appendices H.2–H.6.

**Runtime.** In the scenario of sparse signals and homogeneous lognormal noises, with a bootstrap size $B = 50$, the unconstrained LP scales to $m = 1000$ in ~1.5 minutes per bootstrap replicate, the exact cardinality-constrained MILP becomes the bottleneck for moderate-to-large $m$ (15s at $m = 100$, infeasible at $m \geq 1000$), and Subset-MaxT_FPTAS solves all sizes in under a second, scaling to $m = 5000$ (Appendix H.4).

**Robustness to correlated tests.** In an empirical robustness check, we feed equicorrelated noise $\epsilon_j = \sqrt{\rho}Z + \sqrt{1-\rho}W_j$ (so all pairs have correlation $\rho$) into the independent-bootstrap calibration of SubsetMaxT_FPTAS, and compare against Bonferroni and a correlation-aware Scheffé variant named SchefféCorr (see Appendix H.1). SubsetMaxT_FPTAS retains $\geq$ 99.5% coverage even at $\rho = 0.8$ while remaining substantially tighter than both

baselines; a sign-flipped mixed-correlation design gives nearly identical numbers (full tables in Appendix H.5).

**Sensitivity to bootstrap size $B$.** Empirically, coverage stays at or above the nominal level across all $B$, and the bound mean stabilizes by $B = 50$, with the gap from $B = 50$ to $B = 1000$ under 1.2% (Appendix H.6, Table 20).

## 6.4. Machine Learning Applications

We illustrate the method on ML settings where aggregated effects over a data-selected subset are meaningful. Because ground-truth effects are unknown on real data, we adopt a semi-synthetic protocol to obtain a "ground truth" so that coverage and tightness can be evaluated.

### 6.4.1. SUBGROUP A/B TESTING

We consider a single large randomized experiment with non-overlapping user subgroups (e.g., age bands or regions), so each user belongs to exactly one subgroup and subgroup estimates are independent (Kohavi et al., 2020; Bakshy et al.,

*Table 4.* Synthetic results under dense signals and heterogeneous lognormal noises. Bootstrap size $B = 50$, #replicates $= 200$.

| m | k | Method | Coverage | Bound Mean | Bound SE |
|---|---|--------|----------|------------|----------|
| 30 | 5 | SubsetMaxT | 1.000 | -1.10e-01 | 8.78e-03 |
|  |  | SubsetMaxT_FPTAS | 1.000 | -1.18e-01 | 8.79e-03 |
|  |  | SubsetMaxT(unconstr.) | 1.000 | -1.79e-01 | 9.12e-03 |
|  |  | Scheffé | 1.000 | -1.82e-01 | 9.21e-03 |
|  |  | Bonferroni | 1.000 | -2.46e-01 | 8.30e-03 |
| 30 | 10 | SubsetMaxT | 1.000 | -1.44e-01 | 1.81e-02 |
|  |  | SubsetMaxT_FPTAS | 1.000 | -1.61e-01 | 1.82e-02 |
|  |  | SubsetMaxT(unconstr.) | 1.000 | -1.56e-01 | 1.82e-02 |
|  |  | Scheffé | 1.000 | -3.68e-01 | 1.98e-02 |
|  |  | Bonferroni | 1.000 | -6.96e-01 | 1.94e-02 |
| 30 | 30 | SubsetMaxT | 1.000 | -3.83e-01 | 3.72e-02 |
|  |  | Scheffé | 1.000 | -8.60e-01 | 4.14e-02 |
|  |  | Bonferroni | 1.000 | -1.70e+00 | 4.97e-02 |
| 50 | 5 | SubsetMaxT | 1.000 | -6.75e-02 | 6.93e-03 |
|  |  | SubsetMaxT_FPTAS | 1.000 | -7.41e-02 | 6.93e-03 |
|  |  | SubsetMaxT(unconstr.) | 1.000 | -1.95e-01 | 6.96e-03 |
|  |  | Scheffé | 1.000 | -1.29e-01 | 7.06e-03 |
|  |  | Bonferroni | 1.000 | -1.90e-01 | 6.54e-03 |
| 50 | 10 | SubsetMaxT | 1.000 | -7.81e-02 | 1.33e-02 |
|  |  | SubsetMaxT_FPTAS | 1.000 | -9.39e-02 | 1.33e-02 |
|  |  | SubsetMaxT(unconstr.) | 1.000 | -1.50e-01 | 1.35e-02 |
|  |  | Scheffé | 1.000 | -2.74e-01 | 1.40e-02 |
|  |  | Bonferroni | 1.000 | -5.81e-01 | 1.33e-02 |
| 50 | 50 | SubsetMaxT | 1.000 | -2.30e-01 | 5.30e-02 |
|  |  | Scheffé | 1.000 | -1.12e+00 | 5.67e-02 |
|  |  | Bonferroni | 1.000 | -2.98e+00 | 7.11e-02 |
| 100 | 5 | SubsetMaxT_FPTAS | 1.000 | -7.89e-02 | 6.23e-03 |
|  |  | SubsetMaxT(unconstr.) | 1.000 | -3.56e-01 | 6.49e-03 |
|  |  | Scheffé | 1.000 | -1.23e-01 | 6.15e-03 |
|  |  | Bonferroni | 1.000 | -1.81e-01 | 5.60e-03 |
| 100 | 10 | SubsetMaxT_FPTAS | 1.000 | -4.79e-02 | 1.12e-02 |
|  |  | SubsetMaxT(unconstr.) | 1.000 | -2.96e-01 | 1.10e-02 |
|  |  | Scheffé | 1.000 | -1.82e-01 | 1.09e-02 |
|  |  | Bonferroni | 1.000 | -4.50e-01 | 9.77e-03 |
| 100 | 100 | SubsetMaxT | 1.000 | 7.31e-01 | 5.90e-02 |
|  |  | Scheffé | 1.000 | -8.08e-01 | 6.51e-02 |
|  |  | Bonferroni | 1.000 | -4.74e+00 | 9.45e-02 |

*Table 5.* Hillstrom subgroup A/B testing results. $m = 32$, bootstrap size $B = 50$, #replicates $= 200$.

| k | Method | Coverage | Bound Mean | Bound SE |
|---|--------|----------|------------|----------|
| 5 | SubsetMaxT | 0.995 | 1.37e-02 | 3.34e-04 |
|  | SubsetMaxT(unconstr.) | 0.995 | 1.15e-02 | 3.33e-04 |
|  | Scheffé | 1.000 | 1.17e-02 | 3.28e-04 |
|  | Bonferroni | 1.000 | 9.52e-03 | 3.10e-04 |
| 10 | SubsetMaxT | 0.990 | 3.64e-02 | 7.05e-04 |
|  | SubsetMaxT(unconstr.) | 0.990 | 3.59e-02 | 7.02e-04 |
|  | Scheffé | 0.995 | 3.02e-02 | 6.74e-04 |
|  | Bonferroni | 1.000 | 2.04e-02 | 5.72e-04 |
| 32 | SubsetMaxT | 0.995 | 7.88e-02 | 1.45e-03 |
|  | Scheffé | 1.000 | 6.42e-02 | 1.36e-03 |
|  | Bonferroni | 1.000 | 3.72e-02 | 1.19e-03 |

*Table 6.* Criteo subgroup A/B testing results. $m = 49$, bootstrap size $B = 50$, #replicates $= 200$.

| k | Method | Coverage | Bound Mean | Bound SE |
|---|--------|----------|------------|----------|
| 5 | SubsetMaxT | 1.000 | 6.19e-02 | 7.10e-04 |
|  | SubsetMaxT(unconstr.) | 1.000 | 5.92e-02 | 6.96e-04 |
|  | Scheffé | 1.000 | 6.03e-02 | 7.00e-04 |
|  | Bonferroni | 1.000 | 5.89e-02 | 6.89e-04 |
| 10 | SubsetMaxT | 1.000 | 1.35e-01 | 2.02e-03 |
|  | SubsetMaxT(unconstr.) | 1.000 | 1.33e-01 | 2.01e-03 |
|  | Scheffé | 1.000 | 1.30e-01 | 1.98e-03 |
|  | Bonferroni | 1.000 | 1.25e-01 | 1.93e-03 |
| 49 | SubsetMaxT | 1.000 | 1.40e+00 | 9.40e-03 |
|  | Scheffé | 1.000 | 1.36e+00 | 9.25e-03 |
|  | Bonferroni | 1.000 | 1.30e+00 | 8.92e-03 |

variance of the difference-in-means. We then run subset search and bound construction methods on the $\hat{\theta}_i, \hat{\sigma}_i^2$ for comparison.

**Results summary.** Tables 5–6 summarize results from Hillstrom and Criteo. Across both datasets, SubsetMaxT produces consistently less conservative (larger) lower confidence bounds than Scheffé and Bonferroni while maintaining near-nominal coverage; the gains grow with $k$ on Hillstrom and are smaller but systematic on Criteo. As before, Bonferroni is the most conservative due to the strongest multiplicity correction, Scheffé is intermediate, and SubsetMaxT adapts the critical value to the structured family, yielding tighter post-selection bounds.

### 6.4.2. MULTI-DATASET COMPARISON OF ML PIPELINES

We use tabular benchmarks and select up to 50 datasets from OpenML-CC18. Each dataset is a task $i$, and for each task, we run two pipelines A and B multiple times to compare their performance. Concretely, for each dataset $i$, we run pipeline A and pipeline B across many randomized train/test splits (or cross-validation folds) and record per-run performance differences $d_{ir}$ (e.g., $\text{accuracy}_A - \text{accuracy}_B$). Treating $\{d_{ir}\}_{r=1}^{R_i}$ as the finite population for task $i$, the ground-truth task effect is the finite-population mean $\theta_i^{\text{true}} = \frac{1}{R_i} \sum_r d_{ir}$. To create a controlled semi-synthetic setting, we

2014). We use two public randomized marketing datasets that are standard in uplift/subgroup evaluation: the Hillstrom email campaign dataset and the Criteo uplift prediction dataset (Diemert Eustache, Betlei Artem et al., 2018). We construct non-overlapping subgroups by coarse covariate bins, treat each subgroup as one "test" $i$, and apply the same post-selection bound evaluation protocol. Formally, let $\{Y_{i\ell}^{(1)}\}_{\ell=1}^{N_i^{(1)}}$ and $\{Y_{i\ell}^{(0)}\}_{\ell=1}^{N_i^{(0)}}$ denote observed outcomes for treated and control units in subgroup $i$, respectively. Treating the observed units as a finite population, the ground-truth subgroup effect is

$$\theta_i^{\text{true}} = \frac{1}{N_i^{(1)}} \sum_{\ell=1}^{N_i^{(1)}} Y_{i\ell}^{(1)} - \frac{1}{N_i^{(0)}} \sum_{\ell=1}^{N_i^{(0)}} Y_{i\ell}^{(0)}.$$

To evaluate post-selection inference, we repeatedly resample with replacement within each subgroup and arm, using per-arm resample sizes proportional to the original arm sizes in that subgroup. For each replicate, we set $\hat{\theta}_i$ to the difference in resampled means and estimate $\hat{\sigma}_i^2$ using the plug-in

*Table 7.* Comparison of ML pipelines on $m = 50$ OpenML datasets without injection. Bootstrap size $B = 50$, #replicates $= 200$.

| $k$ | Method | Coverage | Bound Mean | Bound SE |
|---|---|---|---|---|
| 5 | SubsetMaxT | 1.000 | 3.35e-02 | 6.55e-04 |
| | SubsetMaxT(unconstr.) | 1.000 | 3.08e-02 | 6.14e-04 |
| | Scheffé | 1.000 | 3.17e-02 | 6.27e-04 |
| | Bonferroni | 1.000 | 3.02e-02 | 6.05e-04 |
| 10 | SubsetMaxT | 1.000 | 3.92e-02 | 5.95e-04 |
| | SubsetMaxT(unconstr.) | 1.000 | 3.79e-02 | 5.83e-04 |
| | Scheffé | 1.000 | 3.44e-02 | 5.58e-04 |
| | Bonferroni | 1.000 | 2.83e-02 | 5.16e-04 |
| 50 | SubsetMaxT | 1.000 | 3.79e-02 | 5.83e-04 |
| | Scheffé | 1.000 | 3.09e-02 | 5.42e-04 |
| | Bonferroni | 1.000 | 1.91e-02 | 4.78e-04 |

*Table 8.* ML pipeline comparison results with simple injection. $m = 8$, bootstrap size $B = 1000$, #replicates $= 160$.

| $\delta$ | $k$ | Method | Coverage | Bound Mean | p-value |
|---|---|---|---|---|---|
| 0.02 | 1 | SubsetMaxT | 1.000 | -0.007 | - |
| | | Scheffé | 1.000 | -0.007 | 0.025 |
| | | Bonferroni | 1.000 | -0.005 | $< 0.001$ |
| | 8 | SubsetMaxT | 1.000 | -0.018 | - |
| | | Scheffé | 1.000 | -0.022 | 0.134 |
| | | Bonferroni | 1.000 | -0.029 | $< 0.001$ |
| 0.05 | 2 | SubsetMaxT | 1.000 | 0.047 | - |
| | | Scheffé | 1.000 | 0.044 | $< 0.001$ |
| | | Bonferroni | 1.000 | 0.043 | $< 0.001$ |
| | 8 | SubsetMaxT | 1.000 | 0.073 | - |
| | | Scheffé | 1.000 | 0.071 | 0.018 |
| | | Bonferroni | 1.000 | 0.071 | 0.542 |
| 0.08 | 1 | SubsetMaxT | 1.000 | 0.041 | - |
| | | Scheffé | 1.000 | 0.040 | $< 0.001$ |
| | | Bonferroni | 1.000 | 0.043 | $< 0.001$ |
| | 8 | SubsetMaxT | 1.000 | 0.110 | - |
| | | Scheffé | 1.000 | 0.107 | 0.089 |
| | | Bonferroni | 1.000 | 0.105 | $< 0.001$ |

*Table 9.* ML pipeline comparison results under sparse-strong and dense-weak injections. $m = 12$, bootstrap size $B = 10$.

| Pattern | $k$ | Method | Coverage | Bound Mean |
|---|---|---|---|---|
| Sparse-strong | 12 | SubsetMaxT | 0.935 | 0.0735 |
| | | Scheffé | 0.930 | 0.0728 |
| | | Bonferroni | 0.945 | 0.0703 |
| Sparse-strong | 4 | SubsetMaxT | 0.985 | 0.1142 |
| | | Scheffé | 0.995 | 0.1117 |
| | | Bonferroni | 0.995 | 0.1089 |
| Dense-weak | 12 | SubsetMaxT | 0.950 | 0.0467 |
| | | Scheffé | 0.955 | 0.0465 |
| | | Bonferroni | 0.975 | 0.0438 |
| Dense-weak | 4 | SubsetMaxT | 1.000 | 0.0945 |
| | | Scheffé | 1.000 | 0.0922 |
| | | Bonferroni | 1.000 | 0.0897 |

may inject a known shift on a subset of tasks $S^*$ by setting $d'_{ir} = d_{ir} + \delta$ for $i \in S^*$, so $\theta_i^{\text{true,new}} = \theta_i^{\text{true}} + \delta$ for $i \in S^*$. For each replicate, we bootstrap resample with replacement within each task to obtain $(\hat{\theta}_i, \hat{\sigma}_i^2)$, and run subset search and bound construction.

**Results without injection.** Table 7 summarizes the results on 50 datasets from OpenML-CC18 without effect injection comparing two pipelines for classification: k-NN versus random forests. Empirically, all methods attain perfect coverage across the tested cardinality limits. Since we are constructing confidence lower bounds at matched coverage ($\geq 95\%$), a larger "Bound Mean" indicates a tighter bound. Read this way, SubsetMaxT consistently produces the least conservative lower bounds, while Bonferroni incurs the strongest multiplicity penalty and Scheffé is intermediate.

**Results with simple injection.** Table 8 summarizes results on 8 OpenML-CC18 classification datasets spanning diverse domains: cmc (contraceptive method), labor (negotiations), lymph (lymphography), iris (flowers), diabetes (Pima), breast-w (Wisconsin cancer), sonar (mine detection), and balance-scale. Four ML pipelines represent distinct algorithmic families: logistic regression, random forest, k-NN, and naive Bayes. We assign performance boosts $\delta \in \{0.01, 0.02, 0.03, 0.04, 0.05, 0.06, 0.08, 0.10\}$ to randomly selected 60% of datasets. Table 8 shows results averaged over injections. In this experiment, we also include statistical analysis revealing that SubsetMaxT provides statistically significant improvements (paired t-test p-value $< 0.05$ for most conditions). Perfect coverage (100%) is achieved across all methods, and SubsetMaxT delivers tighter bounds than Scheffé and Bonferroni in most settings. The exception is $k = 1$, where each candidate subset is a single test, so the candidate statistics are mutually independent; under such independence Bonferroni's union bound is nearly tight, making it essentially optimal and slightly tighter than SubsetMaxT.

**Results with more sophisticated injection.** Table 9 presents results on 12 OpenML-CC18 classification datasets: kr-vs-kp, letter, mfeat-factors, mfeat-fourier, mfeat-karhunen, mfeat-morphological, mfeat-zernike, cmc, credit-g, diabetes, spambase, and splice. Two ML pipelines compared are logistic regression and random forest. Two shift patterns are tested: sparse-strong (shifting 20% of tasks by $\delta = 0.05$) and dense-weak (shifting 50% of tasks by $\delta = 0.01$). For each task $i$, we estimate the baseline performance gap from 10 repeated train/test splits, then inject known shifts on a subset of tasks and resample to obtain $\hat{\theta}_i$ and $\hat{\sigma}_i$. Results in Table 9 are averaged over 200 injections. The previous observation persists: all three methods achieve the desired coverage while SubsetMaxT delivers tighter bounds than the two baselines.

## Impact Statement

This paper presents work whose goal is to advance the fields of simultaneous inference and its applications to machine learning. There are many potential societal consequences of our work, none of which we feel must be specifically

highlighted here.

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

## A. Proof of Proposition 4.1

*Proof of Proposition 4.1.* For convenience, we drop the $\hat{\boldsymbol{\theta}}$ and $\hat{\Sigma}$ in the notation and simply write $T(S)$ for $T(S; \hat{\theta}, \hat{\Sigma})$. $T(S) > 0$ implies $\sum_{j \in S} \hat{\theta}_j > 0$. Therefore, if $\hat{\theta}_i \leq 0$, we have

$$
\begin{aligned}
T(S \cup \{i\}) &= \frac{\hat{\theta}_i + \sum_{j \in S} \hat{\theta}_j}{\sqrt{\hat{\sigma}_i^2 + \sum_{j \in S} \hat{\sigma}_j^2}} \leq \frac{\sum_{j \in S} \hat{\theta}_j}{\sqrt{\hat{\sigma}_i^2 + \sum_{j \in S} \hat{\sigma}_j^2}} \\
&< \frac{\sum_{j \in S} \hat{\theta}_j}{\sqrt{\sum_{j \in S} \hat{\sigma}_j^2}} = T(S).
\end{aligned}
$$

This proves one direction of the proposition. For the other direction, note that $T(S) < 0$ implies $\sum_{j \in S} \hat{\theta}_j < 0$ and hence if $\hat{\theta}_i \geq 0$ it holds that

$$
\begin{aligned}
T(S \cup \{i\}) &= \frac{\hat{\theta}_i + \sum_{j \in S} \hat{\theta}_j}{\sqrt{\hat{\sigma}_i^2 + \sum_{j \in S} \hat{\sigma}_j^2}} \geq \frac{\sum_{j \in S} \hat{\theta}_j}{\sqrt{\hat{\sigma}_i^2 + \sum_{j \in S} \hat{\sigma}_j^2}} \\
&> \frac{\sum_{j \in S} \hat{\theta}_j}{\sqrt{\sum_{j \in S} \hat{\sigma}_j^2}} = T(S).
\end{aligned}
$$

This completes the proof. $\qquad \square$

## B. Proof of Proposition 4.2

*Proof.* For any non-empty $S \subseteq \{1, 2, \ldots, m\}$, by the definition of $i^*$ we have

$$
\begin{aligned}
\frac{\sum_{i \in S} \hat{\theta}_i}{\sqrt{\sum_{i \in S} \hat{\sigma}_i^2}} &\leq \frac{\sum_{i \in S} \hat{\sigma}_i \cdot \hat{\theta}_{i^*} / \hat{\sigma}_{i^*}}{\sqrt{\sum_{i \in S} \hat{\sigma}_i^2}} \\
&= \frac{\hat{\theta}_{i^*}}{\hat{\sigma}_{i^*}} \frac{\sum_{i \in S} \hat{\sigma}_i}{\sqrt{\sum_{i \in S} \hat{\sigma}_i^2}} \leq \frac{\hat{\theta}_{i^*}}{\hat{\sigma}_{i^*}}.
\end{aligned}
$$

Therefore $T(S; \hat{\theta}, \hat{\Sigma}) \leq T(\{i^*\}; \hat{\theta}, \hat{\Sigma})$. This completes the proof. $\qquad \square$

## C. Proof of Theorem 4.3

*Proof of Theorem 4.3.* Denote by $f$ and $g$ the objective functions of (3) and (4) respectively. Suppose the optimal solutions of (3) and (4) are $\boldsymbol{z}^* = (z_1^*, \ldots, z_m^*)$ and $\boldsymbol{u}^* = (u_1^*, \ldots, u_m^*)$, $\boldsymbol{v}^* = (v_{11}^*, \ldots, v_{mm}^*)$, respectively. Now we prove each of the two directions.

**Reduce (3) to (4).**

Given $\boldsymbol{z}^*$, we define $\boldsymbol{u}, \boldsymbol{v}$ as

$$
u_i = \frac{z_i^*}{\sum_{j \in [m]} \hat{\sigma}_j^2 z_j^*}, v_{ij} = \min\{u_i, u_j\} = \begin{cases} \frac{1}{\sum_{j \in [m]} \hat{\sigma}_j^2 z_j^*} & z_i^* = z_j^* = 1 \\ 0 & else \end{cases}, \forall i, j \in [m].
$$

Direct calculation shows that $(\boldsymbol{u}, \boldsymbol{v})$ is a feasible solution of (4). Then

$$
g(\boldsymbol{v}^*) \geq g(\boldsymbol{v}) = \sum_{i,j \in [m]} \hat{\theta}_i \hat{\theta}_j \frac{z_i^* z_j^*}{\sum_{j \in [m]} \hat{\sigma}_j^2 z_j^*} = f(\boldsymbol{z}^*).
$$

**Reduce (4) to (3).**

Given $\boldsymbol{u}^*, \boldsymbol{v}^*$, now we prove $g(\boldsymbol{v}^*) \leq f(\boldsymbol{z}^*)$. Without loss of generality we assume $v_{ij}^* = \min\{u_i^*, u_j^*\}$.

Define a collection of variables $\boldsymbol{z}(r) = (z_1(r), \ldots, z_m(r))$ indexed by a parameter $r \geq 0$ as

$$z_i(r) := \mathbf{1}\{u_i^* \geq r\}, \forall i \in [m].$$

Then we have

$$\int_0^\infty \sum_{i \in [m]} \hat{\sigma}_i^2 z_i(r) \mathrm{d}r = \sum_{i \in [m]} \hat{\sigma}_i^2 \int_0^\infty \mathbf{1}\{u_i^* \geq r\} \mathrm{d}r = \sum_{i \in [m]} \hat{\sigma}_i^2 u_i^* \leq 1. \tag{6}$$

On the other hand, for $\forall r \geq 0$,

$$\sum_{i,j \in [m]} \hat{\theta}_i \hat{\theta}_j z_i(r) z_j(r) = \sum_{i,j \in [m]} \hat{\theta}_i \hat{\theta}_j \mathbf{1}\{\min\{u_i^*, u_j^*\} \geq r\} = \sum_{i,j \in [m]} \hat{\theta}_i \hat{\theta}_j \mathbf{1}\{v_{ij}^* \geq r\}.$$

Therefore

$$\int_0^\infty \sum_{i,j \in [m]} \hat{\theta}_i \hat{\theta}_j z_i(r) z_j(r) \mathrm{d}r = \int_0^\infty \sum_{i,j \in [m]} \hat{\theta}_i \hat{\theta}_j \mathbf{1}\{v_{ij}^* \geq r\} \mathrm{d}r = \sum_{i,j \in [m]} \hat{\theta}_i \hat{\theta}_j v_{ij}^* = g(\boldsymbol{v}^*). \tag{7}$$

Combining (6) and (7) yields

$$\int_0^\infty \sum_{i,j \in [m]} \hat{\theta}_i \hat{\theta}_j z_i(r) z_j(r) \mathrm{d}r \geq g(\boldsymbol{v}^*) \int_0^\infty \sum_{i \in [m]} \hat{\sigma}_i^2 z_i(r) \mathrm{d}r. \tag{8}$$

Claim: $\exists r \geq 0$ such that $\boldsymbol{z}(r)$ is a feasible solution of (3) (i.e., $\sum_i z_i(r) \geq 1$), and $f(\boldsymbol{z}(r)) \geq g(\boldsymbol{v}^*)$.

Assume for contradiction the claim is false. Then for $\forall r \geq 0$ such that $\sum_i z_i(r) \geq 1$ we have $f(\boldsymbol{z}(r)) < g(\boldsymbol{v}^*)$, which implies that

$$\sum_{i,j \in [m]} \hat{\theta}_i \hat{\theta}_j z_i(r) z_j(r) < g(\boldsymbol{v}^*) \sum_{i \in [m]} \hat{\sigma}_i^2 z_i(r).$$

For $r$ such that $\sum_i z_i(r) = 0$ we have $\sum_{i,j \in [m]} \hat{\theta}_i \hat{\theta}_j z_i(r) z_j(r) = 0 = g(\boldsymbol{v}^*) \sum_{i \in [m]} \hat{\sigma}_i^2 z_i(r)$. Integrating over $r$ yields

$$\int_0^\infty \sum_{i,j \in [m]} \hat{\theta}_i \hat{\theta}_j z_i(r) z_j(r) \mathrm{d}r < g(\boldsymbol{v}^*) \int_0^\infty \sum_{i \in [m]} \hat{\sigma}_i z_i(r) \mathrm{d}r,$$

contradicting (8). The claim is proven, which immediately implies $f(\boldsymbol{z}^*) \geq g(\boldsymbol{v}^*)$.

Combining the two directions, we get $f(\boldsymbol{z}^*) = g(\boldsymbol{v}^*)$, hence the equivalence between (3) and (4). Also note that all possible configurations of $\boldsymbol{z}(r)$ can be efficiently enumerated by setting $r = u_i^*$ for all $i \in [m]$, so optimal solutions of (3) and (4) can be efficiently converted from one to another. $\qquad\square$

## D. Proof of Theorem 4.4

*Proof of Theorem 4.4.* Let $\mathcal{A} = \{\boldsymbol{z} \in \{0,1\}^m : A\boldsymbol{z} \leq b, \mathbf{1}^\top \boldsymbol{z} \geq 1\}$. For any $\boldsymbol{z} \in \mathcal{A}$, define

$$u_i = \frac{z_i}{\sum_{j=1}^m \hat{\sigma}_j^2 z_j}, \qquad v_{ij} = \min\{u_i, u_j\}.$$

Then $(\boldsymbol{u}, \boldsymbol{v}, \boldsymbol{z})$ is feasible for the MILP, and its objective equals

$$\sum_{i,j=1}^m \hat{\theta}_i \hat{\theta}_j v_{ij} = \frac{\sum_{i,j=1}^m \hat{\theta}_i \hat{\theta}_j z_i z_j}{\sum_{j=1}^m \hat{\sigma}_j^2 z_j}.$$

Therefore the MILP optimal value is at least the optimal value of (3).

Conversely, take any feasible MILP solution $(\boldsymbol{u}, \boldsymbol{v}, \boldsymbol{z})$ and let $S = \{i : z_i = 1\}$. Because $A \geq 0$ and $A\boldsymbol{z} \leq \boldsymbol{b}$, every non-empty subset of $S$ is also feasible for $A\boldsymbol{z} \leq \boldsymbol{b}$, so the family is downward-closed. The MILP constraints imply $u_i = 0$ whenever $z_i = 0$, so the support of $(\boldsymbol{u}, \boldsymbol{v})$ lies in $S$. Restricting to indices in $S$ yields a feasible solution of the LP in Theorem 4.3 for the induced problem on $S$ (with $\sum_{i \in S} \hat{\sigma}_i^2 u_i = 1$), hence by Theorem 4.3 the MILP objective is bounded above by the maximum density over non-empty subsets of $S$, so the MILP objective is at most the optimum of (3). This proves equality of optimal values.

Now let $(\boldsymbol{u}^*, \boldsymbol{v}^*, \boldsymbol{z}^*)$ be any optimal MILP solution and let $S^* = \{i : z_i^* = 1\}$. Applying Theorem 4.3 to the LP solution $(\boldsymbol{u}^*, \boldsymbol{v}^*)$ restricted to $S^*$ yields an index $i^*$ such that $z_i := \mathbf{1}\{u_i^* \geq u_{i^*}^*, i \in S^*\}$ attains the same objective value as $(\boldsymbol{u}^*, \boldsymbol{v}^*)$. Let $\boldsymbol{z}$ denote the vector with components $z_i$. Since $\boldsymbol{z}$ is supported on $S^*$ and the family is downward-closed, $\boldsymbol{z}$ is feasible and hence optimal for (3). This proves the claimed conversion from any optimal MILP solution. $\square$

# E. Proof of Theorem 5.1

We consider

$$T(S; \Sigma^{\frac{1}{2}}\hat{\Sigma}^{-\frac{1}{2}}\hat{\boldsymbol{\theta}}, \Sigma) = \frac{\sum_{i \in S} \hat{\theta}_i / \hat{\sigma}_i \cdot \sigma_i}{\sqrt{\sum_{i \in S} \sigma_i^2}}$$

**Lemma E.1.** *Suppose $\hat{\sigma}_i^2 > 0$, $\sigma_i^2 > 0$, and $|\frac{\hat{\sigma}_i^2}{\sigma_i^2} - 1| < \delta$ for all $i$ for some $\delta \in (0, 1)$. Given a non-empty subset $S$, if $\hat{\theta}_i > 0$ for all $i \in S$, then it holds that*

$$\sqrt{\frac{1-\delta}{1+\delta}} T(S; \Sigma^{\frac{1}{2}}\hat{\Sigma}^{-\frac{1}{2}}\hat{\boldsymbol{\theta}}, \Sigma) < T(S; \hat{\boldsymbol{\theta}}, \hat{\Sigma}) < \sqrt{\frac{1+\delta}{1-\delta}} T(S; \Sigma^{\frac{1}{2}}\hat{\Sigma}^{-\frac{1}{2}}\hat{\boldsymbol{\theta}}, \Sigma).$$

*Proof of Lemma E.1.* Fix a non-empty subset $S$. The assumption $|\hat{\sigma}_i^2/\sigma_i^2 - 1| < \delta$ implies $(1-\delta)\sigma_i^2 < \hat{\sigma}_i^2 < (1+\delta)\sigma_i^2$ and hence

$$\sqrt{1-\delta} < \frac{\hat{\sigma}_i}{\sigma_i} < \sqrt{1+\delta}, \qquad \frac{1}{\sqrt{1+\delta}} < \frac{\sigma_i}{\hat{\sigma}_i} < \frac{1}{\sqrt{1-\delta}}.$$

Therefore,

$$\sqrt{1-\delta} \sum_{i \in S} \hat{\theta}_i \frac{\sigma_i}{\hat{\sigma}_i} < \sum_{i \in S} \hat{\theta}_i < \sqrt{1+\delta} \sum_{i \in S} \hat{\theta}_i \frac{\sigma_i}{\hat{\sigma}_i},$$

where we use $\hat{\theta}_i > 0$ for all $i \in S$ to preserve the inequalities after summation. Also,

$$(1-\delta) \sum_{i \in S} \sigma_i^2 < \sum_{i \in S} \hat{\sigma}_i^2 < (1+\delta) \sum_{i \in S} \sigma_i^2.$$

Combining and dividing by the corresponding square roots yields

$$\sqrt{\frac{1-\delta}{1+\delta}} \cdot \frac{\sum_{i \in S} \hat{\theta}_i(\sigma_i/\hat{\sigma}_i)}{\sqrt{\sum_{i \in S} \sigma_i^2}} < \frac{\sum_{i \in S} \hat{\theta}_i}{\sqrt{\sum_{i \in S} \hat{\sigma}_i^2}} < \sqrt{\frac{1+\delta}{1-\delta}} \cdot \frac{\sum_{i \in S} \hat{\theta}_i(\sigma_i/\hat{\sigma}_i)}{\sqrt{\sum_{i \in S} \sigma_i^2}},$$

which is exactly the claimed inequality. $\square$

**Lemma E.2.** *If $\max_{S \in \mathcal{A}} T(S; \hat{\boldsymbol{\theta}}, \hat{\Sigma}) > 0$ and $|\frac{\hat{\sigma}_i^2}{\sigma_i^2} - 1| < \delta$ for all $i$ for some $\delta \in (0, 1)$, we have*

$$\sqrt{\frac{1-\delta}{1+\delta}} \max_{S \in \mathcal{A}} T(S; \Sigma^{\frac{1}{2}}\hat{\Sigma}^{-\frac{1}{2}}\hat{\boldsymbol{\theta}}, \Sigma) < \max_{S \in \mathcal{A}} T(S; \hat{\boldsymbol{\theta}}, \hat{\Sigma}) < \sqrt{\frac{1+\delta}{1-\delta}} \max_{S \in \mathcal{A}} T(S; \Sigma^{\frac{1}{2}}\hat{\Sigma}^{-\frac{1}{2}}\hat{\boldsymbol{\theta}}, \Sigma).$$

*Proof of Lemma E.2.* Let $S^* \in \arg\max_{S \in \mathcal{A}} T(S; \hat{\boldsymbol{\theta}}, \hat{\Sigma})$. The assumption $\max_S T(S; \hat{\boldsymbol{\theta}}, \hat{\Sigma}) > 0$ ensures $T(S^*; \hat{\boldsymbol{\theta}}, \hat{\Sigma}) > 0$, so in particular $\hat{\theta}_i > 0$ for all $i \in S^*$ by Proposition 4.1. By Lemma E.1, we have

$$T(S^*; \hat{\boldsymbol{\theta}}, \hat{\Sigma}) < \sqrt{\frac{1+\delta}{1-\delta}} T(S^*; \Sigma^{\frac{1}{2}}\hat{\Sigma}^{-\frac{1}{2}}\hat{\boldsymbol{\theta}}, \Sigma).$$

Since $T(S^*; \Sigma^{\frac{1}{2}}\hat{\Sigma}^{-\frac{1}{2}}\hat{\boldsymbol{\theta}}, \Sigma) \leq \max_{S \in \mathcal{A}} T(S; \Sigma^{\frac{1}{2}}\hat{\Sigma}^{-\frac{1}{2}}\hat{\boldsymbol{\theta}}, \Sigma)$, the inequality yields the desired upper bound. For the lower bound, consider $S_* \in \arg\max_{S \in \mathcal{A}} T(S; \Sigma^{\frac{1}{2}}\hat{\Sigma}^{-\frac{1}{2}}\hat{\boldsymbol{\theta}}, \Sigma)$, again Lemma E.1 gives

$$\sqrt{\frac{1-\delta}{1+\delta}} T(S_*; \Sigma^{\frac{1}{2}}\hat{\Sigma}^{-\frac{1}{2}}\hat{\boldsymbol{\theta}}, \Sigma) < T(S_*; \hat{\boldsymbol{\theta}}, \hat{\Sigma})$$

and noticing that $T(S_*; \hat{\boldsymbol{\theta}}, \hat{\Sigma}) \leq \max_{S \in \mathcal{A}} T(S; \hat{\boldsymbol{\theta}}, \hat{\Sigma})$ we obtain the lower bound. $\qquad\square$

**Lemma E.3.** *Let $x_1, \ldots, x_B$ and $y_1, \ldots, y_B$ be non-negative numbers such that $(1 - \delta_1)x_b \leq y_b \leq (1 + \delta_2)x_b$ for all $b = 1, \ldots, B$ and some $\delta_1 \in (0, 1)$ and $\delta_2 \in (0, \infty)$. For every $1 \leq b \leq B$, let $x_{(b)}$ and $y_{(b)}$ be their b-th smallest number respectively, then we must have*

$$(1 - \delta_1)x_{(b)} \leq y_{(b)} \leq (1 + \delta_2)x_{(b)}.$$

*Proof of Lemma E.3.* Let $I_b$ be the index set of the $b$ smallest elements of $\{x_i\}$. Since $y_i \leq (1 + \delta_2)x_i$ for all $i$, we have

$$\max_{i \in I_b} y_i \leq (1 + \delta_2)\max_{i \in I_b} x_i = (1 + \delta_2)x_{(b)}.$$

Because $y_{(b)}$ is the $b$-th smallest element of $\{y_i\}$, it cannot exceed the maximum of any $b$ elements, hence $y_{(b)} \leq (1 + \delta_2)x_{(b)}$.

For the lower bound, let $J_b$ be the index set of the $b$ smallest elements of $\{y_i\}$. Since $y_i \geq (1 - \delta_1)x_i$ for all $i$,

$$y_{(b)} = \max_{i \in J_b} y_i \geq (1 - \delta_1)\max_{i \in J_b} x_i.$$

The maximum of any $b$ elements of $\{x_i\}$ is at least $x_{(b)}$, so $\max_{i \in J_b} x_i \geq x_{(b)}$, yielding $y_{(b)} \geq (1 - \delta_1)x_{(b)}$. Combining the two inequalities gives the claim. $\qquad\square$

Consider a coupling of the bootstrap randomness with $\hat{\boldsymbol{\theta}}$ and $\hat{\Sigma}$ that generates the bootstrap samples $\hat{\boldsymbol{\theta}}^b := \hat{\Sigma}^{\frac{1}{2}}\boldsymbol{\omega}_b$, where each $\boldsymbol{\omega}_b$ independently follows the standard normal on $\mathbb{R}^m$ and are also independent from $\hat{\boldsymbol{\theta}}$ and $\hat{\Sigma}$. By construction, the estimated quantile $c_{\text{maxT}}$ is the $\lceil(1 - \alpha)(B + 1)\rceil$-th smallest among

$$\{\max(0, \max_{S \in \mathcal{A}} T(S; \hat{\Sigma}^{\frac{1}{2}}\boldsymbol{\omega}_b, \hat{\Sigma}))\}_{b=1}^B.$$

Denote by $c'_{\text{maxT}}$ the $\lceil(1 - \alpha)(B + 1)\rceil$-th smallest among

$$\{\max(0, \max_{S \in \mathcal{A}} T(S; \Sigma^{\frac{1}{2}}\boldsymbol{\omega}_b, \Sigma))\}_{b=1}^B.$$

**Lemma E.4.** *For every $\boldsymbol{\theta} \in \mathbb{R}^m$ and $\delta \in (0, 1)$, we have*

$$\mathbb{P}\left(\max_{S \in \mathcal{A}} T(S; \hat{\boldsymbol{\theta}} - \boldsymbol{\theta}, \hat{\Sigma}) \leq c_{\text{maxT}}\right) \geq \mathbb{P}\left(\max_{S \in \mathcal{A}} T(S; \Sigma^{\frac{1}{2}}\hat{\Sigma}^{-\frac{1}{2}}(\hat{\boldsymbol{\theta}} - \boldsymbol{\theta}), \Sigma) \leq \frac{1-\delta}{1+\delta}c'_{\text{maxT}}\right) - \mathbb{P}\left(\max_i |\frac{\hat{\sigma}_i^2}{\sigma_i^2} - 1| \geq \delta\right).$$

*Proof of Lemma E.4.* Define the events $E_1 := \{\max_i |\hat{\sigma}_i^2/\sigma_i^2 - 1| < \delta\}$ and

$$E_2 := \left\{\max_{S \in \mathcal{A}} T(S; \Sigma^{\frac{1}{2}}\hat{\Sigma}^{-\frac{1}{2}}(\hat{\boldsymbol{\theta}} - \boldsymbol{\theta}), \Sigma) \leq \frac{1-\delta}{1+\delta}c'_{\text{maxT}}\right\}.$$

Assume $E_1$ and $E_2$ hold. If $\max_{S \in \mathcal{A}} T(S; \hat{\boldsymbol{\theta}} - \boldsymbol{\theta}, \hat{\Sigma}) \leq 0$, then $\max_S T(S; \hat{\boldsymbol{\theta}} - \boldsymbol{\theta}, \hat{\Sigma}) \leq c_{\text{maxT}}$ trivially since $c_{\text{maxT}} \geq 0$ by construction. Otherwise, by Lemma E.2,

$$\max_{S \in \mathcal{A}} T(S; \hat{\boldsymbol{\theta}} - \boldsymbol{\theta}, \hat{\Sigma}) \leq \sqrt{\frac{1+\delta}{1-\delta}} \max_{S \in \mathcal{A}} T(S; \Sigma^{\frac{1}{2}}\hat{\Sigma}^{-\frac{1}{2}}(\hat{\boldsymbol{\theta}} - \boldsymbol{\theta}), \Sigma) \leq \sqrt{\frac{1-\delta}{1+\delta}} c'_{\text{maxT}}.$$

Next, apply Lemma E.2 to the coupled bootstrap samples to get, for each $b$,

$$\max\left(0, \max_{S \in \mathcal{A}} T(S; \hat{\Sigma}^{\frac{1}{2}}\boldsymbol{\omega}_b, \hat{\Sigma})\right) \geq \sqrt{\frac{1-\delta}{1+\delta}} \max\left(0, \max_{S \in \mathcal{A}} T(S; \Sigma^{\frac{1}{2}}\boldsymbol{\omega}_b, \Sigma)\right).$$

By Lemma E.3, the empirical $(1 - \alpha)$ order statistic satisfies

$$c_{\mathrm{maxT}} \geq \sqrt{\frac{1 - \delta}{1 + \delta}}\, c'_{\mathrm{maxT}},$$

so $E_1 \cap E_2 \subseteq \{\max_{S \in \mathcal{A}} T(S; \hat{\boldsymbol{\theta}} - \boldsymbol{\theta}, \hat{\Sigma}) \leq c_{\mathrm{maxT}}\}$.

Therefore,

$$\mathbb{P}\left(\max_{S \in \mathcal{A}} T(S; \hat{\boldsymbol{\theta}} - \boldsymbol{\theta}, \hat{\Sigma}) \leq c_{\mathrm{maxT}}\right) \geq \mathbb{P}(E_1 \cap E_2)$$
$$= \mathbb{P}(E_2) - \mathbb{P}(E_2 \cap E_1^c)$$
$$\geq \mathbb{P}(E_2) - \mathbb{P}(E_1^c),$$

which is exactly the claimed inequality. $\qquad\square$

**Lemma E.5.** *If $(\hat{\theta}_i, \hat{\sigma}_i^2), i = 1, \ldots, m$ are mutually independent, it holds for every $\boldsymbol{\theta} \in \mathbb{R}^m$ and $\delta \in (0, 1)$ that*

$$\left| \mathbb{P}\left(\max_{S \in \mathcal{A}} T(S; \Sigma^{\frac{1}{2}} \hat{\Sigma}^{-\frac{1}{2}}(\hat{\boldsymbol{\theta}} - \boldsymbol{\theta}), \Sigma) \leq \frac{1 - \delta}{1 + \delta} c'_{maxT}\right) - \mathbb{P}\left(\max_{S \in \mathcal{A}} T(S; \Sigma^{\frac{1}{2}} \boldsymbol{\omega}, \Sigma) \leq \frac{1 - \delta}{1 + \delta} c'_{maxT}\right) \right|$$
$$\leq \sum_{i=1}^{m} \sup_{x \in \mathbb{R}} \left| \mathbb{P}\left(\frac{\hat{\theta}_i - \theta_i}{\hat{\sigma}_i} \leq x\right) - \Phi(x) \right|$$

*where $\boldsymbol{\omega}$ is a standard normal on $\mathbb{R}^m$ that is independent from the bootstrap samples, and $\Phi(\cdot)$ is the CDF of the standard normal.*

*Proof of Lemma E.5.* Let $F_i(x) = \mathbb{P}((\hat{\theta}_i - \theta_i)/\hat{\sigma}_i \leq x)$. By independence, the joint law of $\hat{\Sigma}^{-\frac{1}{2}}(\hat{\boldsymbol{\theta}} - \boldsymbol{\theta})$ is the product measure $P_F = \bigotimes_{i=1}^{m} F_i$. Let $P_\Phi = \bigotimes_{i=1}^{m} \Phi$ be the product measure of i.i.d. standard normals. For any fixed threshold $t$, define the set

$$A_t = \left\{ z \in \mathbb{R}^m : \max_{S \in \mathcal{A}} T(S; \Sigma^{\frac{1}{2}} z, \Sigma) \leq t \right\}.$$

Since $\boldsymbol{\omega}$ is independent of the bootstrap randomness, we may condition on $c'_{\mathrm{maxT}}$ and treat $t = \frac{1 - \delta}{1 + \delta} c'_{\mathrm{maxT}}$ as fixed.

Introduce the telescoping sequence of product measures

$$P^{(i)} = \bigotimes_{j=1}^{i} F_j \otimes \bigotimes_{j=i+1}^{m} \Phi, \qquad i = 0, 1, \ldots, m,$$

so $P^{(0)} = P_\Phi$ and $P^{(m)} = P_F$. Then

$$P_F(A_t) - P_\Phi(A_t) = \sum_{i=1}^{m} \left( P^{(i)}(A_t) - P^{(i-1)}(A_t) \right).$$

For each $i$, the difference between $P^{(i)}$ and $P^{(i-1)}$ is only in the $i$-th coordinate. For a vector $z \in \mathbb{R}^m$, let $z_{-i}$ denote the vector of all coordinates except the $i$-th. By definition,

$$T(S; \Sigma^{\frac{1}{2}} z, \Sigma) = \frac{\sum_{j \in S} \sigma_j z_j}{\sqrt{\sum_{j \in S} \sigma_j^2}},$$

so the event $A_t$ can be written as

$$A_t = \bigcap_{S \in \mathcal{A}} \left\{ z : \sum_{j \in S} \sigma_j z_j \leq t \sqrt{\sum_{j \in S} \sigma_j^2} \right\}.$$

Hence for any fixed $z_{-i}$, the section

$$A_t(z_{-i}) = \{z_i : (z_i, z_{-i}) \in A_t\}$$

is an interval in the form of $(-\infty, x]$. Therefore,

$$\left| \int \mathbf{1}_{A_t(z_{-i})}(z_i)\, dF_i(z_i) - \int \mathbf{1}_{A_t(z_{-i})}(z_i)\, d\Phi(z_i) \right| \leq \sup_{x \in \mathbb{R}} |F_i(x) - \Phi(x)|.$$

Integrating over $z_{-i}$ with respect to the remaining coordinates yields

$$\left| P^{(i)}(A_t) - P^{(i-1)}(A_t) \right| \leq \sup_{x \in \mathbb{R}} |F_i(x) - \Phi(x)|.$$

Summing over $i$ yields

$$\left| P_F(A_t) - P_\Phi(A_t) \right| \leq \sum_{i=1}^{m} \sup_{x \in \mathbb{R}} |F_i(x) - \Phi(x)|.$$

Finally, noting that this bound holds true for all $t$ and taking expectation over $c'_{\text{maxT}}$ yields the stated inequality. $\qquad\square$

**Lemma E.6** (Expected bootstrap quantile bound). *For every $\varepsilon \in (0, \alpha)$,*

$$\mathbb{E}[c'_{maxT}] \leq \Phi^{-1}\left( 1 - \frac{\alpha - \varepsilon}{|\mathcal{A}|} \right) \;+\; \sqrt{2e^{-2B\varepsilon^2}}\, \sqrt{2\log(B|\mathcal{A}|) + 1},$$

*where $\Phi$ is the standard normal CDF.*

*Proof of Lemma E.6.* Denote by $M := \max_{S \in \mathcal{A}} T(S; \Sigma^{\frac{1}{2}}\boldsymbol{\omega}, \Sigma)$ where $\boldsymbol{\omega} \sim N(0, I_m)$, and let $F$ be the CDF of $\max(0, M)$. For any $t > 0$, $\mathbb{P}(\max(0, M) > t) = \mathbb{P}(M > t)$, so the same upper-tail bounds apply. Since each $T(S; \Sigma^{\frac{1}{2}}\boldsymbol{\omega}, \Sigma)$ follows the standard normal, we have

$$\mathbb{P}(M > t) = \mathbb{P}\left( \bigcup_{S \in \mathcal{A}} \{T(S; \Sigma^{\frac{1}{2}}\boldsymbol{\omega}, \Sigma) > t\} \right) \leq \sum_{S \in \mathcal{A}} \mathbb{P}(T(S; \Sigma^{\frac{1}{2}}\boldsymbol{\omega}, \Sigma) > t) = |\mathcal{A}|\bar{\Phi}(t).$$

Thus for any $\beta \in (0, 1)$ the $(1 - \beta)$-quantile $q_{1-\beta}$ of $M$ satisfies

$$q_{1-\beta} \leq \Phi^{-1}\left( 1 - \frac{\beta}{|\mathcal{A}|} \right). \tag{9}$$

Let $M_{(b)}$ denote the empirical $(1 - \alpha)$ order statistic from $B$ i.i.d. draws $M_1, \ldots, M_B$ of $M$ and $F_B$ denote the empirical CDF of the $B$ draws. Given an $\varepsilon \in (0, \alpha)$, the Dvoretzky–Kiefer–Wolfowitz inequality states that

$$\mathbb{P}\left( \sup_{x \in \mathbb{R}} |F_B(x) - F(x)| > \varepsilon \right) \leq 2e^{-2B\varepsilon^2}.$$

Denote by $q = F^{-1}(1 - \alpha + \varepsilon)$. Since $M_{(b)} > q$ implies $F_B(q) < 1 - \alpha$, we have

$$|F_B(q) - F(q)| > F(q) - (1 - \alpha) = \varepsilon,$$

and therefore

$$\mathbb{P}(M_{(b)} > q) \leq \mathbb{P}\left( \sup_{x \in \mathbb{R}} |F_B(x) - F(x)| > \varepsilon \right) \leq 2e^{-2B\varepsilon^2}.$$

Then

$$\mathbb{E}[M_{(b)}] \leq q + \mathbb{E}[M_{(B)}\mathbf{1}\{M_{(b)} > q\}] \leq q + \sqrt{\mathbb{E}[M_{(B)}^2]}\sqrt{2e^{-2B\varepsilon^2}},$$

where $M_{(B)} = \max_{1 \leq b \leq B} M_b$ and we used Cauchy–Schwarz. Using (9) with $\beta = \alpha - \varepsilon$ gives

$$q \leq \Phi^{-1}\left( 1 - \frac{\alpha - \varepsilon}{|\mathcal{A}|} \right),$$

which yields the stated bound. It remains to bound $\mathbb{E}[M_{(B)}^2]$. Since

$$M_{(B)} = \max_{1 \leq b \leq B} \max_{S \in \mathcal{A}} T(S; \Sigma^{\frac{1}{2}}\boldsymbol{\omega}_b, \Sigma),$$

with $T(S; \Sigma^{\frac{1}{2}} \boldsymbol{\omega}_b, \Sigma) \sim N(0, 1)$. By a union bound,

$$\mathbb{P}(M_{(B)} > t) \leq B|\mathcal{A}| \, \bar{\Phi}(t).$$

Using $\bar{\Phi}(t) \leq \frac{\phi(t)}{t}$ for $t > 0$ and the tail-integral formula,

$$\mathbb{E}[M_{(B)}^2] = \int_0^\infty \mathbb{P}(M_{(B)} > \sqrt{s}) \, ds \leq t_0^2 + \int_{t_0}^\infty 2t \cdot B|\mathcal{A}| \, \bar{\Phi}(t) \, dt \leq t_0^2 + 2B|\mathcal{A}| \int_{t_0}^\infty \phi(t) \, dt,$$

where $t_0 > 0$ is arbitrary. Choosing $t_0 = \sqrt{2 \log(B|\mathcal{A}|)}$ yields

$$\mathbb{E}[M_{(B)}^2] \leq 2 \log(B|\mathcal{A}|) + 2B|\mathcal{A}| \, \bar{\Phi}\big(\sqrt{2 \log(B|\mathcal{A}|)}\big) \leq 2 \log(B|\mathcal{A}|) + 1,$$

since $B|\mathcal{A}| \, \bar{\Phi}(\sqrt{2 \log(B|\mathcal{A}|)}) \leq B|\mathcal{A}| \, \frac{\phi(\sqrt{2 \log(B|\mathcal{A}|)})}{\sqrt{2 \log(B|\mathcal{A}|)}} \leq \frac{1}{\sqrt{2\pi}}$. $\qquad \square$

**Lemma E.7** (Anti-concentration). *There exists a universal constant $C_{\mathrm{ac}} > 0$ such that, for any $\delta \in (0, 1)$,*

$$\mathbb{P}\left(\max_{S \in \mathcal{A}} T(S; \Sigma^{\frac{1}{2}} \boldsymbol{\omega}, \Sigma) \leq \frac{1 - \delta}{1 + \delta} c'_{maxT}\right)$$
$$\geq \mathbb{P}\left(\max_{S \in \mathcal{A}} T(S; \Sigma^{\frac{1}{2}} \boldsymbol{\omega}, \Sigma) \leq c'_{maxT}\right) - C_{\mathrm{ac}} \cdot \frac{2\delta}{1 + \delta} \, \mathbb{E}[c'_{maxT}] \cdot \big(\sqrt{\log(|\mathcal{A}|)} + 1\big).$$

*Proof of Lemma E.7.* Let $M := \max_{S \in \mathcal{A}} T(S; \Sigma^{\frac{1}{2}} \boldsymbol{\omega}, \Sigma)$. Condition on $c'_{\mathrm{maxT}}$ and set $\varepsilon := c'_{\mathrm{maxT}} - \frac{1-\delta}{1+\delta} c'_{\mathrm{maxT}} = \frac{2\delta}{1+\delta} c'_{\mathrm{maxT}}$. Then

$$\mathbb{P}(M \leq c'_{\mathrm{maxT}} - \varepsilon \,|\, c'_{\mathrm{maxT}}) = \mathbb{P}(M \leq c'_{\mathrm{maxT}} \,|\, c'_{\mathrm{maxT}}) - \mathbb{P}(M \in (c'_{\mathrm{maxT}} - \varepsilon, \, c'_{\mathrm{maxT}}] \,|\, c'_{\mathrm{maxT}}).$$

By the Gaussian anti-concentration inequality such as Nazarov's inequality (Chernozhukov et al., 2017), there exists $C_{\mathrm{ac}} > 0$ such that

$$\sup_{t \in \mathbb{R}} \mathbb{P}\big(M \in [t, t + \varepsilon]\big) \leq C_{\mathrm{ac}} \varepsilon \big(\sqrt{\log(|\mathcal{A}|)} + 1\big).$$

Taking expectations over $c'_{\mathrm{maxT}}$ on both sides yields

$$\mathbb{P}\left(M \leq \frac{1 - \delta}{1 + \delta} c'_{\mathrm{maxT}}\right) \geq \mathbb{P}(M \leq c'_{\mathrm{maxT}}) - C_{\mathrm{ac}} \cdot \frac{2\delta}{1 + \delta} \, \mathbb{E}[c'_{\mathrm{maxT}}] \cdot \big(\sqrt{\log(|\mathcal{A}|)} + 1\big),$$

which is the stated bound. $\qquad \square$

Now we are ready to prove the theorem.

*Proof of Theorem 5.1.* By Lemma E.4,

$$\mathbb{P}\left(\max_{S \in \mathcal{A}} T(S; \hat{\boldsymbol{\theta}} - \boldsymbol{\theta}, \hat{\Sigma}) \leq c_{\mathrm{maxT}}\right) \geq \mathbb{P}\left(\max_{S \in \mathcal{A}} T(S; \Sigma^{\frac{1}{2}} \hat{\Sigma}^{-\frac{1}{2}} (\hat{\boldsymbol{\theta}} - \boldsymbol{\theta}), \Sigma) \leq \frac{1 - \delta}{1 + \delta} c'_{\mathrm{maxT}}\right)$$
$$- \mathbb{P}\left(\max_i \left|\frac{\hat{\sigma}_i^2}{\sigma_i^2} - 1\right| \geq \delta\right). \tag{10}$$

Applying Lemma E.5 yields

$$\mathbb{P}\left(\max_{S \in \mathcal{A}} T(S; \Sigma^{\frac{1}{2}} \hat{\Sigma}^{-\frac{1}{2}} (\hat{\boldsymbol{\theta}} - \boldsymbol{\theta}), \Sigma) \leq \frac{1 - \delta}{1 + \delta} c'_{\mathrm{maxT}}\right) \geq \mathbb{P}\left(\max_{S \in \mathcal{A}} T(S; \Sigma^{\frac{1}{2}} \boldsymbol{\omega}, \Sigma) \leq \frac{1 - \delta}{1 + \delta} c'_{\mathrm{maxT}}\right)$$
$$- \sum_{i=1}^m \sup_{x \in \mathbb{R}} \left|\mathbb{P}\left(\frac{\hat{\theta}_i - \theta_i}{\hat{\sigma}_i} \leq x\right) - \Phi(x)\right|. \tag{11}$$

Let $N := |\mathcal{A}|$. Lemma E.7 then gives

$$\mathbb{P}\left(\max_{S \in \mathcal{A}} T(S; \Sigma^{\frac{1}{2}}\boldsymbol{\omega}, \Sigma) \leq \frac{1-\delta}{1+\delta}c'_{\text{maxT}}\right) \geq \mathbb{P}\left(\max_{S \in \mathcal{A}} T(S; \Sigma^{\frac{1}{2}}\boldsymbol{\omega}, \Sigma) \leq c'_{\text{maxT}}\right)$$
$$- C_{\text{ac}} \cdot \frac{2\delta}{1+\delta}\, \mathbb{E}[c'_{\text{maxT}}] \cdot \left(\sqrt{\log(N)} + 1\right). \tag{12}$$

By Lemma E.6,

$$\mathbb{E}[c'_{\text{maxT}}] \leq \Phi^{-1}\left(1 - \frac{\alpha - \varepsilon}{N}\right) + \sqrt{2e^{-2B\varepsilon^2}}\sqrt{2\log(BN) + 1}.$$

Let $\varepsilon = \alpha/\sqrt{2}$. Using $\Phi^{-1}(1-x) \leq \sqrt{2\log(1/x)}$ for $x \in (0, 1/2)$ and $\alpha - \varepsilon = \alpha(1 - 1/\sqrt{2})$, we have

$$\Phi^{-1}\left(1 - \frac{\alpha - \varepsilon}{N}\right) \leq C\sqrt{\log(N/\alpha)}.$$

Therefore

$$\mathbb{E}[c'_{\text{maxT}}](\sqrt{\log(N)} + 1) \leq C\sqrt{\log(N/\alpha)}\sqrt{\log N} + Ce^{-B\alpha^2/2}\sqrt{\log(BN)}\sqrt{\log N}$$
$$\leq C\log(N/\alpha) + Ce^{-B\alpha^2/2}\sqrt{\log(BN)\log N}.$$

Since $\log(BN) = \log B + \log N$, we have

$$\sqrt{\log(BN)\log N} \leq \sqrt{\log B \,\log N} + \log N$$

and hence

$$C_{\text{ac}} \cdot \frac{2\delta}{1+\delta}\, \mathbb{E}[c'_{\text{maxT}}] \cdot \left(\sqrt{\log(N)} + 1\right) \leq C\delta\left(\log(N/\alpha) + e^{-B\alpha^2/2}\sqrt{\log(B)\log(N)}\right). \tag{13}$$

Finally, since $c'_{\text{maxT}}$ is the $\lceil(1-\alpha)(B+1)\rceil$-th order statistic of $B$ i.i.d. draws from $\max(0, M)$, we have

$$\mathbb{P}\left(\max_{S \in \mathcal{A}} T(S; \Sigma^{\frac{1}{2}}\boldsymbol{\omega}, \Sigma) \leq c'_{\text{maxT}}\right) = \mathbb{P}\left(\max(0, \max_{S \in \mathcal{A}} T(S; \Sigma^{\frac{1}{2}}\boldsymbol{\omega}, \Sigma)) \leq c'_{\text{maxT}}\right) \geq \frac{\lceil(1-\alpha)(B+1)\rceil}{B+1} \geq 1 - \alpha. \tag{14}$$

Combining the inequalities (10)(11)(12)(13)(14) yields

$$\mathbb{P}\left(\max_{S \in \mathcal{A}} T(S; \hat{\boldsymbol{\theta}} - \boldsymbol{\theta}, \hat{\Sigma}) \leq c_{\text{maxT}}\right) \geq 1 - \alpha - \mathcal{E},$$

where

$$\mathcal{E} = \sum_{i=1}^{m} \sup_{x \in \mathbb{R}} \left|\mathbb{P}\left(\frac{\hat{\theta}_i - \theta_i}{\hat{\sigma}_i} \leq x\right) - \Phi(x)\right|$$
$$+ \mathbb{P}\left(\max_i \left|\frac{\hat{\sigma}_i^2}{\sigma_i^2} - 1\right| \geq \delta\right) + C\delta\left(\log(|\mathcal{A}|/\alpha) + e^{-B\alpha^2/2}\sqrt{\log(B)\log(|\mathcal{A}|)}\right),$$

for every $\delta \in (0, 1)$. Taking the infimum over $\delta$ gives the conclusion. $\qquad\square$

# F. Proof of Corollary 5.2

*Proof of Corollary 5.2.* We bound the two components of $\mathcal{E}$ in Theorem 5.1 under the stated assumptions. For the first component, the Berry–Esseen condition gives $\sup_x |\mathbb{P}((\hat{\theta}_i - \theta_i)/\hat{\sigma}_i \leq x) - \Phi(x)| \leq C_{\text{BE}}/\sqrt{n_i}$ for each $i$, so

$$\sum_{i=1}^{m} \sup_{x \in \mathbb{R}} \left|\mathbb{P}\left(\frac{\hat{\theta}_i - \theta_i}{\hat{\sigma}_i} \leq x\right) - \Phi(x)\right| \leq C_{\text{BE}} \sum_{i=1}^{m} \frac{1}{\sqrt{n_i}}.$$

For the second component, a union bound and the variance-concentration condition give, for every $\delta \in (0,1)$,

$$\mathbb{P}\left(\max_i \left|\frac{\hat{\sigma}_i^2}{\sigma_i^2} - 1\right| \geq \delta\right) \leq \sum_{i=1}^m \mathbb{P}\left(\left|\frac{\hat{\sigma}_i^2}{\sigma_i^2} - 1\right| \geq \delta\right) \leq \sum_{i=1}^m 2e^{-c\,n_i\delta^2} \leq 2m\,e^{-c\,n_{\min}\delta^2},$$

using $n_i \geq n_{\min}$. Writing $L = \log(|\mathcal{A}|/\alpha) + e^{-B\alpha^2/2}\sqrt{\log(B)\log(|\mathcal{A}|)}$ and taking the infimum over $\delta$ yields the first claimed bound,

$$\mathcal{E} \leq C_{\mathrm{BE}} \sum_{i=1}^m \frac{1}{\sqrt{n_i}} + \inf_{\delta \in (0,1)} \left[2m\,e^{-c\,n_{\min}\delta^2} + C\,\delta\,L\right].$$

For the explicit bound, suppose $n_{\min} > \log(mn_{\min})/c$ and take $\delta = \sqrt{\log(mn_{\min})/(c\,n_{\min})}$, which then lies in $(0,1)$. With this choice $c\,n_{\min}\delta^2 = \log(mn_{\min})$, so $2m\,e^{-c\,n_{\min}\delta^2} = 2m/(mn_{\min}) = 2/n_{\min}$ and $C\,\delta\,L = C\,L\sqrt{\log(mn_{\min})/(c\,n_{\min})}$. Substituting into the infimum bound gives

$$\mathcal{E} \leq C_{\mathrm{BE}} \sum_{i=1}^m \frac{1}{\sqrt{n_i}} + \frac{2}{n_{\min}} + C\,L\sqrt{\frac{\log(mn_{\min})}{c\,n_{\min}}},$$

as claimed. $\qquad\qquad\qquad\qquad\qquad\qquad\qquad\qquad\qquad\qquad\qquad\qquad\qquad\qquad\qquad\qquad\quad\square$

## G. Detailed FPTAS for Constrained Families

This section provides the full algorithm and proof for the FPTAS announced in Section 4.4. After sign-screening (Proposition 4.1), we restrict attention to $\hat{\theta}_i > 0$ and study the squared problem

$$T^{*2} = \max_{1 \leq |S| \leq k} \frac{P(S)^2}{Q(S)}, \qquad P(S) := \sum_{i \in S} \hat{\theta}_i, \quad Q(S) := \sum_{i \in S} \hat{\sigma}_i^2.$$

Let $a_i := \hat{\theta}_i > 0$ and $b_i := \hat{\sigma}_i^2 > 0$. The crucial structural observation is that the objective depends on $\boldsymbol{z} \in \{0,1\}^m$ only through the two linear aggregates $P(S)$ and $Q(S)$. This 2D sufficient statistic enables a dynamic-programming algorithm that discretizes $Q$ and tracks $P$ exactly. The construction is the natural ratio-objective analogue of the classical knapsack FPTAS (Ibarra & Kim, 1975).

**Algorithm.** Fix $\epsilon \in (0,1)$. Sort $b_{(1)} \geq b_{(2)} \geq \cdots \geq b_{(m)}$ and let $b_{\min} = \min_i b_i$, $Q_{\mathrm{ub}}^{(k)} = \sum_{i=1}^k b_{(i)}$. Enumerate geometric guesses

$$\hat{Q} \in \{b_{\min}, 2b_{\min}, 4b_{\min}, \ldots, Q_{\mathrm{ub}}^{(k)}\},$$

a list of length $L_Q = O(\log(k \cdot b_{\max}/b_{\min}))$. Initialize $\widetilde{T} \leftarrow 0$. For each guess $\hat{Q}$:

1. **Filter and discretize.** Exclude items with $b_i > 2\hat{Q}$; let $I_{\hat{Q}} = \{i : b_i \leq 2\hat{Q}\}$ and $k_{\hat{Q}} = \min(k, |I_{\hat{Q}}|)$. Set $\delta_Q = \epsilon\hat{Q}/k_{\hat{Q}}$ and discretize by ceiling, $\bar{b}_i = \lceil b_i/\delta_Q \rceil$. Since each retained item satisfies $\bar{b}_i = \lceil b_i/\delta_Q \rceil \leq \lceil 2\hat{Q}/\delta_Q \rceil = \lceil 2k_{\hat{Q}}/\epsilon \rceil$ and at most $k_{\hat{Q}}$ items are selected, the maximum reachable discretized aggregate is $\bar{Q}_{\max} = k_{\hat{Q}}\lceil 2k_{\hat{Q}}/\epsilon \rceil = O(k_{\hat{Q}}^2/\epsilon)$.

2. **DP table.** Relabel the elements of $I_{\hat{Q}}$ as $1, 2, \ldots, |I_{\hat{Q}}|$ and write $[j] := \{1, \ldots, j\}$ for its first $j$ elements. For $j \in \{0, 1, \ldots, |I_{\hat{Q}}|\}$, $c \in \{0, 1, \ldots, k_{\hat{Q}}\}$, and $\bar{q} \in \{0, 1, \ldots, \bar{Q}_{\max}\}$, define

$$H[j][c][\bar{q}] = \max\left\{\sum_{i \in S} a_i : S \subseteq [j], \ |S| = c, \ \sum_{i \in S} \bar{b}_i = \bar{q}\right\},$$

with $H[0][0][0] = 0$ and $-\infty$ otherwise. The transition adds item $j$:

$$H[j][c][\bar{q}] = \max\left(H[j-1][c][\bar{q}], \ H[j-1][c-1][\bar{q}-\bar{b}_j] + a_j\right).$$

3. **Update best.** Compute $R_{\hat{Q}} = \max_{c \in [k_{\hat{Q}}], \bar{q} \geq 1: H[|I_{\hat{Q}}|][c][\bar{q}] > -\infty} \frac{H[|I_{\hat{Q}}|][c][\bar{q}]}{\sqrt{\bar{q} \cdot \delta_Q}}$ and update $\widetilde{T} \leftarrow \max(\widetilde{T}, R_{\hat{Q}})$.

Return $\widetilde{T}$. Only the optimal value is needed for the bootstrap calibration, so the table can be rolled over $j$ (updating $c$ and $\bar{q}$ in decreasing order) to use only $O(k_{\hat{Q}}\bar{Q}_{\max}) = O(k^3/\epsilon)$ memory; we therefore do not attempt to recover the attaining subset.

**Theorem G.1** (FPTAS approximation guarantee). *The algorithm above returns $\widetilde{T}$ satisfying $T^*/\sqrt{1+\epsilon} \leq \widetilde{T} \leq T^*$, hence $(1-\epsilon)T^* \leq \widetilde{T} \leq T^*$, in time $O(mk^3 L_Q/\epsilon)$.*

Theorem G.1 is the formal version of Theorem 4.5 stated in the main text: it exhibits the algorithm achieving the claimed $(1-\epsilon)T^* \leq \widetilde{T} \leq T^*$ guarantee, and its running time $O(mk^3 L_Q/\epsilon)$ is exactly the $O(mk^3/\epsilon)$ of Theorem 4.5 up to the logarithmic factor $L_Q$ in the input encoding length.

*Proof.* Let $S^*$ be an optimal solution with $P^* = P(S^*)$, $Q^* = Q(S^*)$, $s^* = |S^*| \leq k$, so $T^* = P^*/\sqrt{Q^*}$. Since $b_i > 0$, we have $b_{\min} \leq Q^* \leq Q_{\mathrm{ub}}^{(k)}$, so some geometric guess $\hat{Q}$ satisfies $Q^*/2 \leq \hat{Q} \leq Q^*$.

*Exclusion safety.* For any $i \in S^*$, $b_i \leq Q^* \leq 2\hat{Q}$, so $S^* \subseteq I_{\hat{Q}}$ and $s^* \leq k_{\hat{Q}}$.

*DP correctness.* By induction on $j$ (standard knapsack DP argument), $H[|I_{\hat{Q}}|][c][\bar{q}]$ equals the maximum of $\sum_{i \in S} a_i$ over all $S \subseteq I_{\hat{Q}}$ with $|S| = c$ and $\sum_{i \in S} \bar{b}_i = \bar{q}$.

*Approximation bound.* The subset $S^*$ is feasible in the DP, so $H[|I_{\hat{Q}}|][s^*][\bar{Q}^*] \geq P^*$, where $\bar{Q}^* := \sum_{i \in S^*} \bar{b}_i$. Ceiling rounding gives $\bar{b}_i \delta_Q \geq b_i$, hence $\bar{Q}^* \cdot \delta_Q \geq Q^*$; also $\bar{Q}^* \cdot \delta_Q \leq Q^* + s^* \delta_Q \leq Q^* + \epsilon\hat{Q} \leq Q^* + \epsilon Q^* = (1+\epsilon)Q^*$. Therefore

$$R_{\hat{Q}} \geq \frac{P^*}{\sqrt{\bar{Q}^* \cdot \delta_Q}} \geq \frac{P^*}{\sqrt{(1+\epsilon)Q^*}} = \frac{T^*}{\sqrt{1+\epsilon}}.$$

Since $\widetilde{T} \geq R_{\hat{Q}}$ for this guess, $\widetilde{T} \geq T^*/\sqrt{1+\epsilon} \geq (1-\epsilon)T^*$.

Conversely, every value compared in the "Update best" step has the form $H[|I_{\hat{Q}}|][c][\bar{q}]/\sqrt{\bar{q}\,\delta_Q}$, and by DP correctness $H[|I_{\hat{Q}}|][c][\bar{q}] = \sum_{i \in S} a_i = P(S)$ for some feasible $S \subseteq I_{\hat{Q}}$ with $|S| = c \leq k$ and $\sum_{i \in S} \bar{b}_i = \bar{q}$. Ceiling rounding gives $\bar{q}\,\delta_Q = \sum_{i \in S} \bar{b}_i \delta_Q \geq \sum_{i \in S} b_i = Q(S)$, so $H[|I_{\hat{Q}}|][c][\bar{q}]/\sqrt{\bar{q}\,\delta_Q} = P(S)/\sqrt{\bar{q}\,\delta_Q} \leq P(S)/\sqrt{Q(S)} = T(S) \leq T^*$. Hence $R_{\hat{Q}} \leq T^*$ for every guess, and therefore $\widetilde{T} \leq T^*$. Combining the two bounds gives $(1-\epsilon)T^* \leq \widetilde{T} \leq T^*$.

*Time complexity.* For each guess, the DP table has $O(|I_{\hat{Q}}| \cdot k_{\hat{Q}} \cdot \bar{Q}_{\max}) = O(mk^3/\epsilon)$ entries (using $|I_{\hat{Q}}| \leq m$, $k_{\hat{Q}} \leq k$, and $\bar{Q}_{\max} = O(k^2/\epsilon)$), each filled with $O(1)$ work. Summing over the $L_Q$ geometric guesses gives total running time $O(mk^3 L_Q/\epsilon)$, polynomial in the input encoding length. $\square$

*Remark* G.2 (A symmetric variant). Symmetrically, one can instead discretize the numerator aggregate $P$ and track the denominator aggregate $Q$ exactly, now minimizing $Q$ over subsets reaching each discretized $P$ level. The choice between the two variants is governed by the dynamic ranges $a_{\max}/a_{\min}$ vs $b_{\max}/b_{\min}$; in experiments we select the variant adaptively, discretizing whichever aggregate has the smaller dynamic range and tracking the other exactly.

*Remark* G.3 (Single weighted constraint). We expect a similar approach to extend to a single downward-closed weight constraint $\sum_i w_i z_i \leq W$ with $w_i \geq 0$, for instance by discretizing both $P$ and $Q$ while tracking the constraint weight $\sum_i w_i z_i$ exactly so that feasibility is preserved. We do not pursue a formal treatment of this case here.

Our restriction to a single weight constraint is not incidental: it tracks the boundary of FPTAS-tractability for the underlying knapsack structure. The cardinality bound together with a single weight constraint is the analogue of the classical $0/1$ knapsack, which admits an FPTAS (Ibarra & Kim, 1975). With two or more independent linear constraints, however, the problem becomes a multidimensional (vector) knapsack, which admits a PTAS for any fixed number of constraints but no FPTAS unless P = NP (Korte & Schrader, 1981): each additional constraint introduces a dimension whose rounding may violate feasibility, precluding an error that is polynomially controllable in $1/\epsilon$. This is why we restrict attention to the cardinality constraint together with at most a single weight constraint.

**Rescaling for valid bootstrap calibration.** The FPTAS output underestimates $T^*$, so plugging $\widetilde{T}$ directly into Algorithm 1 as the per-replicate bootstrap maximum would underestimate the critical value $c_{\mathrm{maxT}}$ and risk undercoverage. To preserve validity, rescale each bootstrap statistic by $1/(1-\epsilon)$:

$$\widehat{T} := \widetilde{T}/(1-\epsilon) \geq T^*,$$

so $\widehat{T}$ is a valid upper bound on the per-replicate true subset-maximum. The resulting critical value is inflated by at most a multiplicative factor $1/(1-\epsilon)$, giving an $O(\epsilon)$ width increase. In our experiments we use $\epsilon = 0.01$, corresponding to less than 1% width inflation.

# H. Additional Experimental Details

## H.1. Baseline Methods and Subset Search Algorithm

We summarize the baseline confidence-bound constructions used in experiments.

**Scheffé Bounds.** In our setting the tests are independent with (possibly different) variances, so we take $\Sigma = \mathrm{diag}(\sigma_1^2, \ldots, \sigma_m^2)$. We only need nonnegative contrasts (subset sums), i.e., $a \geq 0$. If the variances are known, then for any $a \geq 0$,

$$T(a) = \frac{a^\top (\hat{\boldsymbol{\theta}} - \boldsymbol{\theta})}{\sqrt{a^\top \Sigma a}} \sim N(0, 1).$$

By Cauchy–Schwarz with the positive part $x_+ := (\max\{x_1, 0\}, \ldots, \max\{x_m, 0\})$,

$$\sup_{a \geq 0} \frac{a^\top (\hat{\boldsymbol{\theta}} - \boldsymbol{\theta})}{\sqrt{a^\top \Sigma a}} = \sqrt{(\hat{\boldsymbol{\theta}} - \boldsymbol{\theta})_+^\top \Sigma^{-1} (\hat{\boldsymbol{\theta}} - \boldsymbol{\theta})_+}.$$

If $\hat{\boldsymbol{\theta}} - \boldsymbol{\theta}$ is component-wise nonpositive, then the supremum is

$$\max_{1 \leq i \leq m} \frac{\hat{\theta}_i - \theta_i}{\sigma_i} < 0,$$

so the positive-part expression above is an upper bound in that case. Let $c_{\mathrm{Sch}}$ be the $(1 - \alpha)$ quantile of the right-hand side under $N(0, \Sigma)$; this can be computed by Monte Carlo using $Z \sim N(0, \hat{\Sigma})$ and $s = \sqrt{Z_+^\top \hat{\Sigma}^{-1} Z_+}$. Then with probability $1 - \alpha$,

$$a^\top (\hat{\boldsymbol{\theta}} - \boldsymbol{\theta}) \leq c_{\mathrm{Sch}} \sqrt{a^\top \Sigma a} \qquad \text{for all } a \geq 0.$$

In practice we only observe per-test variance estimates $\hat{\sigma}_i^2$; we therefore use the plug-in version with $\hat{\Sigma} = \mathrm{diag}(\hat{\sigma}_1^2, \ldots, \hat{\sigma}_m^2)$ and the same $c_{\mathrm{Sch}}$ (asymptotically valid under consistent variance estimation). For subset sums (take $a$ as the indicator of $S$), this yields the baseline

$$L_{\mathrm{Sch}}(S) = \sum_{i \in S} \hat{\theta}_i - c_{\mathrm{Sch}} \sqrt{\sum_{i \in S} \hat{\sigma}_i^2}.$$

**Bonferroni Bounds.** Bonferroni treats each subset statistic as a separate test. Let $N = |\mathcal{A}|$ be the number of admissible subsets; a Bonferroni threshold is

$$c_{\mathrm{Bon}} = \Phi^{-1}\left(1 - \frac{\alpha}{N}\right),$$

and the lower bound is

$$L_{\mathrm{Bon}}(S) = \sum_{i \in S} \hat{\theta}_i - c_{\mathrm{Bon}} \sqrt{\sum_{i \in S} \hat{\sigma}_i^2}.$$

This yields valid simultaneous bounds but becomes extremely conservative when $N$ is large.

**Correlation-aware Scheffé bound.** The Scheffé construction above plugs in $\hat{\Sigma} = \mathrm{diag}(\hat{\sigma}_1^2, \ldots, \hat{\sigma}_m^2)$ and is valid under independence. For a general (possibly non-diagonal) covariance matrix $C$, a correlation-aware variant—used in Section 6.3 as SchefféCorr—uses the chi-squared critical value $c_{\mathrm{SchCorr}} = \sqrt{\chi_{m, 1-\alpha}^2}$ paired with the joint standard error $\sqrt{e_S^\top C e_S}$, giving the bound

$$L_{\mathrm{SchCorr}}(S) = \sum_{i \in S} \hat{\theta}_i - \sqrt{\chi_{m, 1-\alpha}^2} \sqrt{e_S^\top C e_S}.$$

This follows from $(\hat{\boldsymbol{\theta}} - \boldsymbol{\theta})^\top C^{-1} (\hat{\boldsymbol{\theta}} - \boldsymbol{\theta}) \sim \chi_m^2$ and Cauchy–Schwarz, and remains valid under any positive-definite $C$ when $C$ is known (or consistently estimated).

*Table 10.* Synthetic results under sparse signals and homogeneous normal noises.

| m | k | Method | Coverage | Bound Mean | Bound SE |
|---|---|---|---|---|---|
| 30 | 5 | SubsetMaxT | 1.000 | 1.33e+00 | 1.96e-02 |
| | | SubsetMaxT(unconstr.) | 1.000 | 1.25e+00 | 1.95e-02 |
| | | Scheffé | 1.000 | 1.29e+00 | 1.94e-02 |
| | | Bonferroni | 1.000 | 1.22e+00 | 1.92e-02 |
| 30 | 10 | SubsetMaxT | 1.000 | 1.27e+00 | 2.01e-02 |
| | | SubsetMaxT(unconstr.) | 1.000 | 1.26e+00 | 2.01e-02 |
| | | Scheffé | 1.000 | 1.18e+00 | 1.95e-02 |
| | | Bonferroni | 1.000 | 9.95e-01 | 1.88e-02 |
| 30 | 30 | SubsetMaxT | 1.000 | 1.26e+00 | 2.01e-02 |
| | | Scheffé | 1.000 | 1.15e+00 | 1.95e-02 |
| | | Bonferroni | 1.000 | 8.86e-01 | 1.84e-02 |
| 50 | 5 | SubsetMaxT | 1.000 | 2.15e+00 | 2.30e-02 |
| | | SubsetMaxT(unconstr.) | 1.000 | 1.88e+00 | 2.31e-02 |
| | | Scheffé | 1.000 | 2.11e+00 | 2.30e-02 |
| | | Bonferroni | 1.000 | 1.99e+00 | 2.24e-02 |
| 50 | 10 | SubsetMaxT | 1.000 | 2.01e+00 | 2.41e-02 |
| | | SubsetMaxT(unconstr.) | 1.000 | 1.94e+00 | 2.41e-02 |
| | | Scheffé | 1.000 | 1.92e+00 | 2.39e-02 |
| | | Bonferroni | 1.000 | 1.62e+00 | 2.29e-02 |
| 50 | 50 | SubsetMaxT | 1.000 | 1.94e+00 | 2.41e-02 |
| | | Scheffé | 1.000 | 1.76e+00 | 2.39e-02 |
| | | Bonferroni | 1.000 | 1.16e+00 | 2.28e-02 |
| 100 | 5 | SubsetMaxT(unconstr.) | 1.000 | 4.09e+00 | 2.78e-02 |
| | | Scheffé | 1.000 | 4.71e+00 | 2.76e-02 |
| | | Bonferroni | 1.000 | 4.59e+00 | 2.71e-02 |
| 100 | 10 | SubsetMaxT(unconstr.) | 1.000 | 6.26e+00 | 4.69e-02 |
| | | Scheffé | 1.000 | 6.63e+00 | 4.79e-02 |
| | | Bonferroni | 1.000 | 6.21e+00 | 4.60e-02 |
| 100 | 100 | SubsetMaxT | 1.000 | 6.28e+00 | 4.66e-02 |
| | | Scheffé | 1.000 | 5.94e+00 | 4.58e-02 |
| | | Bonferroni | 1.000 | 4.59e+00 | 4.16e-02 |

**Subset search algorithms.** The coverage of SubsetMaxT (Theorem 5.1) holds regardless of which subset search algorithm is used—the bootstrap critical value $c_{\mathrm{maxT}}$ protects against the worst-case subset in $\mathcal{A}$. Different search algorithms therefore affect only which subset is reported (and hence the bound's value at that subset). We use three searches in our experiments:

- **GLRScanSearch** (default): A greedy mean-shift likelihood-ratio scan, standard in the scan-statistics literature (Neill, 2012; Kulldorff, 1997). In the heteroskedastic Gaussian setting, a one-sided mean-shift likelihood ratio for a subset $S$ yields the score

$$\mathrm{Score}(S) = \max_{\mu>0}\left\{\mu\sum_{i\in S}\frac{\hat{\theta}_i}{\hat{\sigma}_i^2} - \frac{\mu^2}{2}\sum_{i\in S}\frac{1}{\hat{\sigma}_i^2}\right\} = \frac{\left(\sum_{i\in S}\hat{\theta}_i/\hat{\sigma}_i^2\right)_+^2}{2\sum_{i\in S}1/\hat{\sigma}_i^2}.$$

  Items are sorted by the priority $\hat{\theta}_i/\hat{\sigma}_i^2$ and the best feasible prefix is returned (under a cardinality constraint, only prefixes satisfying it are evaluated).

- **TstatOrderSearch**: Items are sorted by the per-test $t$-statistic $t_i = \hat{\theta}_i/\hat{\sigma}_i$ and the best feasible prefix is returned.

- **ForwardStepwiseSearch**: A greedy forward selection that at each step adds the item maximizing $T(S\cup\{j\};\hat{\boldsymbol{\theta}},\hat{\Sigma})$ among feasible additions.

## H.2. Runtime

Table 14 reports per-bootstrap-replicate runtimes for the exact LP (unconstrained, $k = m$), the exact MILP ($k = 5$), and SubsetMaxT_FPTAS ($k = 5$, $\epsilon = 0.01$) under sparse signals and homogeneous lognormal noises with $B = 50$.

## H.3. Subset Search Algorithm Comparison

We compare the three subset search algorithms (TstatOrderSearch, ForwardStepwiseSearch, GLRScanSearch) under sparse signals and heterogeneous lognormal noises. Coverage is 1.000 across all settings, as guaranteed by Theorem 5.1; the tables

*Table 11.* Synthetic results under sparse signals and heterogeneous normal noises.

| m | k | Method | Coverage | Bound Mean | Bound SE |
|---|---|---|---|---|---|
| 30 | 5 | SubsetMaxT | 1.000 | 2.21e+00 | 1.51e-02 |
| | | SubsetMaxT(unconstr.) | 1.000 | 2.15e+00 | 1.45e-02 |
| | | Scheffé | 1.000 | 2.16e+00 | 1.45e-02 |
| | | Bonferroni | 1.000 | 2.10e+00 | 1.38e-02 |
| 30 | 10 | SubsetMaxT | 1.000 | 2.16e+00 | 1.46e-02 |
| | | SubsetMaxT(unconstr.) | 1.000 | 2.15e+00 | 1.45e-02 |
| | | Scheffé | 1.000 | 2.07e+00 | 1.36e-02 |
| | | Bonferroni | 1.000 | 1.93e+00 | 1.26e-02 |
| 30 | 30 | SubsetMaxT | 1.000 | 2.15e+00 | 1.45e-02 |
| | | Scheffé | 1.000 | 2.05e+00 | 1.36e-02 |
| | | Bonferroni | 1.000 | 1.85e+00 | 1.26e-02 |
| 50 | 5 | SubsetMaxT | 1.000 | 1.56e+00 | 3.59e-02 |
| | | SubsetMaxT(unconstr.) | 1.000 | 1.36e+00 | 3.31e-02 |
| | | Scheffé | 1.000 | 1.48e+00 | 3.47e-02 |
| | | Bonferroni | 1.000 | 1.38e+00 | 3.31e-02 |
| 50 | 10 | SubsetMaxT | 1.000 | 1.47e+00 | 3.54e-02 |
| | | SubsetMaxT(unconstr.) | 1.000 | 1.41e+00 | 3.44e-02 |
| | | Scheffé | 1.000 | 1.33e+00 | 3.33e-02 |
| | | Bonferroni | 1.000 | 1.08e+00 | 2.95e-02 |
| 50 | 50 | SubsetMaxT | 1.000 | 1.41e+00 | 3.41e-02 |
| | | Scheffé | 1.000 | 1.19e+00 | 3.15e-02 |
| | | Bonferroni | 1.000 | 6.76e-01 | 2.89e-02 |
| 100 | 5 | SubsetMaxT(unconstr.) | 1.000 | 2.52e+00 | 2.82e-02 |
| | | Scheffé | 1.000 | 2.95e+00 | 2.88e-02 |
| | | Bonferroni | 1.000 | 2.85e+00 | 2.83e-02 |
| 100 | 10 | SubsetMaxT(unconstr.) | 1.000 | 4.50e+00 | 6.21e-02 |
| | | Scheffé | 1.000 | 4.69e+00 | 6.45e-02 |
| | | Bonferroni | 1.000 | 4.31e+00 | 6.04e-02 |
| 100 | 100 | SubsetMaxT | 1.000 | 4.61e+00 | 6.59e-02 |
| | | Scheffé | 1.000 | 4.15e+00 | 6.15e-02 |
| | | Bonferroni | 1.000 | 2.90e+00 | 4.96e-02 |

below report only bound means.

Across both search algorithms and all sizes (Tables 15–16), SubsetMaxT and SubsetMaxT_FPTAS produce the tightest (largest) bounds. The relative gap to Scheffé and Bonferroni persists as $m$ grows from 30 to 5,000, confirming that the FPTAS variant remains effective at large scale and that the choice of search algorithm does not affect coverage validity (consistent with Theorem 5.1).

### H.4. Large-$m$ SubsetMaxT_FPTAS vs. Baselines

Table 17 reports SubsetMaxT_FPTAS vs. baselines under dense signals and homogeneous lognormal noises at large $m$, complementing the synthetic Tables 1–4 which target moderate $m \leq 100$. SubsetMaxT_FPTAS consistently produces the tightest bounds while remaining computationally feasible.

### H.5. Robustness to Correlated Tests

To assess SubsetMaxT's empirical robustness when the independence assumption is violated, we generate test statistics with controllable inter-test correlation under two designs.

**Data generation.** Let $W_1, \ldots, W_m$ and $Z$ be i.i.d. $N(0,1)$. For the positive equicorrelation design, we set $\epsilon_j = \sqrt{\rho}Z + \sqrt{1-\rho}W_j$, giving $\text{Cov}(\epsilon_i, \epsilon_j) = \rho$ for all $i \neq j$. For the sign-flipped mixed design, we take the same one-factor noise and flip the signs of half the components, yielding correlation $+\rho$ within each half and $-\rho$ across halves. The corresponding observed test statistics are $\hat{\theta}_j = \theta_j + s_j \epsilon_j / \sqrt{n}$ with $n = 50$ per test, where the per-test scales $s_j > 0$ are drawn from a lognormal distribution to induce heterogeneous variances; the equicorrelation $\rho$ is preserved while the marginal variances $s_j^2/n$ differ across tests. All experiments use $m = 100$, $k = 5$, sparse signals with heterogeneous noise levels, and $B = 50$ with 200 replicates.

Across both designs (Tables 18–19), SubsetMaxT_FPTAS maintains coverage at or above 99.5% (well above the nominal

*Table 12.* Synthetic results under dense signals and homogeneous normal noises.

| m | k | Method | Coverage | Bound Mean | Bound SE |
|---|---|---|---|---|---|
| 30 | 5 | SubsetMaxT | 0.965 | 8.48e-02 | 1.50e-02 |
| | | SubsetMaxT(unconstr.) | 0.990 | -2.43e-02 | 1.50e-02 |
| | | Scheffé | 0.975 | 2.83e-02 | 1.49e-02 |
| | | Bonferroni | 0.990 | -6.55e-02 | 1.40e-02 |
| 30 | 10 | SubsetMaxT | 0.980 | 2.56e-01 | 2.36e-02 |
| | | SubsetMaxT(unconstr.) | 0.980 | 2.44e-01 | 2.37e-02 |
| | | Scheffé | 0.995 | 1.05e-01 | 2.36e-02 |
| | | Bonferroni | 1.000 | -2.40e-01 | 2.23e-02 |
| 30 | 30 | SubsetMaxT | 0.985 | 3.49e-01 | 2.98e-02 |
| | | Scheffé | 1.000 | 1.24e-01 | 2.94e-02 |
| | | Bonferroni | 1.000 | -4.51e-01 | 2.76e-02 |
| 50 | 5 | SubsetMaxT | 0.955 | 1.19e-01 | 1.58e-02 |
| | | SubsetMaxT(unconstr.) | 1.000 | -1.40e-01 | 1.67e-02 |
| | | Scheffé | 0.975 | 7.32e-02 | 1.60e-02 |
| | | Bonferroni | 1.000 | -3.69e-02 | 1.54e-02 |
| 50 | 10 | SubsetMaxT | 0.960 | 3.62e-01 | 2.44e-02 |
| | | SubsetMaxT(unconstr.) | 0.985 | 2.66e-01 | 2.49e-02 |
| | | Scheffé | 0.990 | 2.40e-01 | 2.43e-02 |
| | | Bonferroni | 1.000 | -1.63e-01 | 2.36e-02 |
| 50 | 50 | SubsetMaxT | 0.980 | 8.58e-01 | 4.40e-02 |
| | | Scheffé | 1.000 | 4.82e-01 | 4.31e-02 |
| | | Bonferroni | 1.000 | -7.35e-01 | 4.10e-02 |
| 100 | 5 | SubsetMaxT(unconstr.) | 1.000 | -4.42e-01 | 1.33e-02 |
| | | Scheffé | 0.970 | 1.35e-01 | 1.33e-02 |
| | | Bonferroni | 1.000 | 2.51e-02 | 1.24e-02 |
| 100 | 10 | SubsetMaxT(unconstr.) | 1.000 | -5.82e-03 | 1.98e-02 |
| | | Scheffé | 0.995 | 3.63e-01 | 2.01e-02 |
| | | Bonferroni | 1.000 | -5.46e-02 | 1.92e-02 |
| 100 | 100 | SubsetMaxT | 0.995 | 2.02e+00 | 5.41e-02 |
| | | Scheffé | 1.000 | 1.31e+00 | 5.24e-02 |
| | | Bonferroni | 1.000 | -1.54e+00 | 4.75e-02 |

95%) even at $\rho = 0.8$, while delivering bounds that are 2–4× tighter than SchefféCorr at moderate-to-large $\rho$. Bonferroni is robust to correlation (does not use the covariance) but remains uniformly more conservative than SubsetMaxT_FPTAS. Positive and mixed designs produce nearly identical SubsetMaxT_FPTAS bounds, suggesting the sign structure of correlations has little impact on the method.

### H.6. Bootstrap-Size Sensitivity

Table 20 reports bound mean across $B \in \{30, 50, 200, 500, 1000\}$ for both the exact LP-based SubsetMaxT (at $k = m = 100$) and the constrained SubsetMaxT_FPTAS ($k = 5$, $\epsilon = 0.01$) under sparse signals and heterogeneous lognormal noises. The bound mean stabilizes by $B \approx 50$ in both cases; increasing $B$ to 1000 shrinks the bound by less than 1.2% for SubsetMaxT and less than 2.2% for SubsetMaxT_FPTAS.

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

*Table 13.* Synthetic results under dense signals and heterogeneous normal noises.

| m | k | Method | Coverage | Bound Mean | Bound SE |
|---|---|---|---|---|---|
| 30 | 5 | SubsetMaxT | 0.985 | 4.12e-02 | 1.17e-02 |
| | | SubsetMaxT(unconstr.) | 1.000 | -3.25e-02 | 1.17e-02 |
| | | Scheffé | 0.995 | -2.78e-02 | 1.19e-02 |
| | | Bonferroni | 0.995 | -9.52e-02 | 1.09e-02 |
| 30 | 10 | SubsetMaxT | 0.990 | 1.53e-01 | 2.19e-02 |
| | | SubsetMaxT(unconstr.) | 0.990 | 1.41e-01 | 2.21e-02 |
| | | Scheffé | 1.000 | -5.02e-02 | 2.23e-02 |
| | | Bonferroni | 1.000 | -3.73e-01 | 2.15e-02 |
| 30 | 30 | SubsetMaxT | 1.000 | 1.17e-01 | 3.69e-02 |
| | | Scheffé | 1.000 | -2.74e-01 | 3.79e-02 |
| | | Bonferroni | 1.000 | -1.01e+00 | 3.89e-02 |
| 50 | 5 | SubsetMaxT | 1.000 | 5.48e-02 | 1.02e-02 |
| | | SubsetMaxT(unconstr.) | 1.000 | -8.64e-02 | 1.05e-02 |
| | | Scheffé | 1.000 | -3.29e-03 | 1.03e-02 |
| | | Bonferroni | 1.000 | -6.89e-02 | 1.00e-02 |
| 50 | 10 | SubsetMaxT | 0.995 | 2.35e-01 | 1.99e-02 |
| | | SubsetMaxT(unconstr.) | 0.995 | 1.59e-01 | 2.03e-02 |
| | | Scheffé | 1.000 | 5.38e-02 | 2.00e-02 |
| | | Bonferroni | 1.000 | -2.61e-01 | 1.93e-02 |
| 50 | 50 | SubsetMaxT | 0.990 | 5.61e-01 | 5.35e-02 |
| | | Scheffé | 1.000 | -1.45e-01 | 5.40e-02 |
| | | Bonferroni | 1.000 | -1.73e+00 | 5.78e-02 |
| 100 | 5 | SubsetMaxT(unconstr.) | 1.000 | -2.17e-01 | 9.61e-03 |
| | | Scheffé | 1.000 | 5.72e-02 | 9.40e-03 |
| | | Bonferroni | 1.000 | -6.94e-03 | 8.80e-03 |
| 100 | 10 | SubsetMaxT(unconstr.) | 1.000 | 6.70e-02 | 1.70e-02 |
| | | Scheffé | 1.000 | 2.11e-01 | 1.65e-02 |
| | | Bonferroni | 1.000 | -7.24e-02 | 1.55e-02 |
| 100 | 100 | SubsetMaxT | 0.995 | 2.00e+00 | 5.76e-02 |
| | | Scheffé | 1.000 | 7.45e-01 | 5.63e-02 |
| | | Bonferroni | 1.000 | -2.69e+00 | 6.63e-02 |

*Table 14.* Per-bootstrap-replicate runtime (seconds), sparse signals and homogeneous lognormal noises. "—" marks settings where the MILP is not solved within a reasonable time budget.

| $m$ | LP ($k = m$) | MILP ($k = 5$) | SubsetMaxT_FPTAS ($k = 5$) |
|---|---|---|---|
| 30 | 0.07 | 0.32 | 0.03 |
| 50 | 0.14 | 1.82 | 0.04 |
| 100 | 0.45 | 15.9 | 0.04 |
| 500 | 10.9 | — | 0.15 |
| 1,000 | 89 | — | 0.26 |
| 5,000 | $3.9 \times 10^4$ | — | 1.18 |

*Table 15.* Bound means under TstatOrderSearch. Sparse signals and heterogeneous lognormal noises, $B = 50$.

| $m$ | $k$ | Method | Bound Mean |
|---|---|---|---|
| 30 | 5 | SubsetMaxT | 2.105 |
| | | SubsetMaxT_FPTAS | 2.100 |
| | | Scheffé | 2.059 |
| | | Bonferroni | 2.019 |
| 50 | 10 | SubsetMaxT | 1.763 |
| | | SubsetMaxT_FPTAS | 1.748 |
| | | Scheffé | 1.583 |
| | | Bonferroni | 1.301 |
| 100 | 10 | SubsetMaxT_FPTAS | 5.924 |
| | | Scheffé | 5.694 |
| | | Bonferroni | 5.232 |
| 500 | 10 | SubsetMaxT_FPTAS | 9.438 |
| | | Scheffé | 9.309 |
| | | Bonferroni | 9.053 |
| 1,000 | 10 | SubsetMaxT_FPTAS | 11.224 |
| | | Scheffé | 11.123 |
| | | Bonferroni | 10.920 |
| 5,000 | 10 | SubsetMaxT_FPTAS | 11.130 |
| | | Scheffé | 11.074 |
| | | Bonferroni | 10.965 |

*Table 16.* Bound means under ForwardStepwiseSearch. Sparse signals and heterogeneous lognormal noises, $B = 50$.

| $m$ | $k$ | Method | Bound Mean |
|---|---|---|---|
| 30 | 5 | SubsetMaxT | 1.891 |
| | | SubsetMaxT_FPTAS | 1.887 |
| | | Scheffé | 1.851 |
| | | Bonferroni | 1.816 |
| 50 | 10 | SubsetMaxT | 1.379 |
| | | SubsetMaxT_FPTAS | 1.369 |
| | | Scheffé | 1.254 |
| | | Bonferroni | 1.056 |
| 100 | 10 | SubsetMaxT_FPTAS | 4.629 |
| | | Scheffé | 4.464 |
| | | Bonferroni | 4.134 |
| 500 | 10 | SubsetMaxT_FPTAS | 6.936 |
| | | Scheffé | 6.842 |
| | | Bonferroni | 6.658 |
| 1,000 | 10 | SubsetMaxT_FPTAS | 8.707 |
| | | Scheffé | 8.629 |
| | | Bonferroni | 8.472 |
| 5,000 | 10 | SubsetMaxT_FPTAS | 10.031 |
| | | Scheffé | 9.981 |
| | | Bonferroni | 9.883 |

*Table 17.* SubsetMaxT_FPTAS vs. baselines at large $m$, dense signals and homogeneous lognormal noises, $B = 50$.

| $m$ | $k$ | Method | Coverage | Bound Mean | Runtime (s) |
|---|---|---|---|---|---|
| 500 | 5 | SubsetMaxT_FPTAS | 1.00 | 0.124 | 0.150 |
| | | Scheffé | 1.00 | 0.116 | 0.001 |
| | | Bonferroni | 1.00 | 0.051 | $< 0.001$ |
| 1,000 | 10 | SubsetMaxT_FPTAS | 1.00 | 0.323 | 0.670 |
| | | Scheffé | 1.00 | 0.309 | 0.001 |
| | | Bonferroni | 1.00 | 0.089 | $< 0.001$ |
| 5,000 | 10 | SubsetMaxT_FPTAS | 1.00 | 0.232 | 3.407 |
| | | Scheffé | 1.00 | 0.222 | 0.005 |
| | | Bonferroni | 1.00 | 0.014 | $< 0.001$ |

*Table 18.* Robustness to positive equicorrelation. SubsetMaxT_FPTAS uses the independent-bootstrap calibration despite the correlated data; SchefféCorr uses the true covariance.

| $\rho$ | SubsetMaxT_FPTAS | | SchefféCorr | Bonferroni |
|---|---|---|---|---|
| | Coverage | Bound Mean | Bound Mean | Bound Mean |
| 0.0 | 1.000 | 3.017 | 1.405 | 2.847 |
| 0.1 | 1.000 | 3.000 | 0.893 | 2.831 |
| 0.2 | 1.000 | 2.991 | 0.442 | 2.821 |
| 0.5 | 0.995 | 2.969 | $-0.683$ | 2.798 |
| 0.8 | 0.995 | 2.960 | $-1.597$ | 2.787 |

*Table 19.* Robustness to sign-flipped mixed correlation ($+\rho$ within halves, $-\rho$ across halves).

| $\rho$ | SubsetMaxT_FPTAS | | SchefféCorr | Bonferroni |
|---|---|---|---|---|
| | Coverage | Bound Mean | Bound Mean | Bound Mean |
| 0.1 | 1.000 | 2.994 | 0.891 | 2.825 |
| 0.2 | 1.000 | 2.982 | 0.441 | 2.813 |
| 0.5 | 0.995 | 2.965 | $-0.673$ | 2.794 |
| 0.8 | 0.995 | 2.954 | $-1.580$ | 2.781 |

*Table 20.* Sensitivity of bound mean to bootstrap size $B$. Sparse signals and heterogeneous lognormal noises, $m = 100$, #replicates $= 200$. All coverages are 1.00.

| $B$ | SubsetMaxT (LP, $k = 100$) | SubsetMaxT_FPTAS ($k = 5$) |
|---|---|---|
| 30 | 5.182 | 2.990 |
| 50 | 5.243 | 3.018 |
| 200 | 5.298 | 3.053 |
| 500 | 5.305 | 3.054 |
| 1,000 | 5.303 | 3.055 |

