# OpenReview forum: "Simultaneous Confidence Bounds for Aggregated Effects via Exact Subset Optimization"
_ICML.cc/2026/Conference — ICML 2026 regular_

### Official Review · Reviewer_APZ1 · 2026-03-11

**Soundness:** 2
**Presentation:** 2
**Significance:** 2
**Originality:** 3
**Overall Recommendation:** 4
**Confidence:** 4

**Summary:**

This paper studies the problem of post-selection inference for data-driven chosen subsets in multiple testing scenarios. The authors propose a method for constructing simultaneous confidence bounds. This is achieved by transforming the problem of maximizing the normalized aggregated statistic into a weighted densest-subgraph problem, solving it exactly using Linear Programming (LP) and Mixed-Integer Linear Programming (MILP), and then calibrating the bounds via multiplier bootstrap to obtain valid confidence lower bounds for data-selected subsets. The theoretical part provides finite-sample coverage guarantees, and the experimental section validates the method's effectiveness on synthetic data, subgroup A/B testing, and multi-dataset ML pipeline comparisons.

**Compliance With Llm Reviewing Policy:**

Affirmed.

**Final Justification:**

The rebuttal meaningfully addressed my main concerns, so although some limitations remain, it increased my confidence in the paper’s technical soundness and practical relevance enough to raise my recommendation to Weak Accept.

**Key Questions For Authors:**

1. Limitations of the Independence Assumption: In most ML applications, tests (e.g., features, tasks, subgroups) exhibit complex correlations. Can your method remain valid in the presence of such dependencies? If not, are there plans to extend the theory to handle correlations? Alternatively, are there empirical guidelines for practitioners (e.g., pre-whitening the data)?
2. Computational Scalability: Could you please provide actual runtimes for the MILP solver for different values of $m$ (e.g., $m = 100, 500, 1000, 5000$) and for different bootstrap sizes $B$? For a bootstrap loop with $B=1000$, is the total runtime acceptable (e.g., < 24 hours) when $m > 1000$? If computational cost is prohibitive, are there potential acceleration strategies (e.g., parallelization, approximate algorithms that still retain some theoretical guarantees)?
3. Limitations of Experimental Scale: The maximum $m$ in the experiments is 100, which is far from the scales encountered in many ML problems (e.g., tens of thousands of genes in genomics, millions of items in recommendation systems). Do your small-scale experiments provide any indication of the method's potential effectiveness for larger-scale problems? Are there simulations or theoretical analyses suggesting scalability to higher dimensions?
4. Connection to Cutting-Edge ML Applications: Beyond A/B testing and algorithm comparison, can you envision applications of this method in more cutting-edge ML domains, such as assessing the trustworthiness of large language models, testing cross-modal consistency in multi-modal learning, or comparing policies in reinforcement learning? Addressing this could enhance the paper's appeal to the ICML community.

**Limitations:**

The paper does not sufficiently discuss the following limitations:

- The strong restriction imposed by the independence assumption and its frequent violation in ML applications.
- The computational cost and scalability issues associated with MILP.
- The inapplicability to subset families that are not downward-closed (e.g., connected subgraphs, path constraints).
- The gap between the small experimental scale ($m \leq 100$) and the scales of real-world ML problems.
- The justification for choosing $B=50$ for bootstrap and its impact on the robustness of the results.

It is recommended to add a dedicated "Discussion of Limitations" section in the final version, honestly addressing these points and exploring potential mitigation strategies or future work directions.

**Strengths And Weaknesses:**

Strengths:
1. The paper presents a clear and technically sound framework, with the main theoretical results developed in a coherent and convincing way. In particular, the sign-screening property and the LP/MILP reformulation are both intuitive and well motivated.
2. Reformulating the subset optimization problem as a weighted densest-subgraph problem is a novel and elegant idea. This perspective connects post-selection inference with exact optimization in a principled way.
3. The empirical evaluation covers several relevant settings, including synthetic data, subgroup A/B testing, and ML pipeline comparison. The semi-synthetic design is also reasonable, since it allows coverage and tightness to be evaluated against known ground truth.

Weaknesses
1. The theoretical guarantees rely heavily on an independence assumption that is too restrictive for many modern ML settings, where correlations among tests are common. The paper does not provide either theoretical extensions or empirical robustness checks under dependence.
2. The computational scalability of the method remains unclear. Since the bootstrap procedure repeatedly calls the LP/MILP solver, the overall cost could become substantial, but runtime and solver behavior are not reported in enough detail.
3. The experiments are conducted only at relatively small scales, with m up to around 100. This makes it difficult to judge whether the method would remain practical in larger and more realistic ML problems.
4. The choice of bootstrap size is not sufficiently justified. Although the theory includes a finite-B error term, the paper does not provide practical guidance or a sensitivity analysis showing how performance changes with B.

---

> ### Author Rebuttal · Authors · 2026-03-31
>
> We thank the reviewer for the detailed and constructive feedback. We address each concern below.
>
> ## Independence Assumption
>
> The independence assumption is not an artificial restriction but a natural feature of many important applications. Independent test statistics arise routinely in practice:
>
> - **A/B testing with non-overlapping subgroups:** Subgroup-specific treatment effect estimates are exactly independent when subgroups partition the population (each unit belongs to exactly one subgroup). This is the standard setup in online experimentation (Kohavi, R., Tang, D. & Xu, Y., *Trustworthy Online Controlled Experiments: A Practical Guide to A/B Testing*, Cambridge University Press, 2020).
> - **Meta-analysis:** Meta-analysis combines results from multiple independent studies to estimate an overall effect. Each study is conducted independently with distinct participants, making study-level effect estimates independent by construction — the standard assumption in the meta-analysis literature (Hedges, L. V. & Olkin, I., *Statistical Methods for Meta-Analysis*, Academic Press, 2014; Borenstein, M. et al., *Introduction to Meta-Analysis*, Wiley, 2009).
> - **ML benchmarking across datasets:** Performance estimates on separate benchmark datasets are independent (Demšar, J., "Statistical Comparisons of Classifiers over Multiple Data Sets," *JMLR* 7:1–30, 2006) — this is exactly the setting of our ML pipeline experiments.
>
> These examples cover several important application domains — clinical trials, online experimentation, meta-analysis, and ML evaluation — where independence is a natural structural feature of how data is collected, not an artificial restriction.
>
> That said, we acknowledge that correlation among tests is common in other applications (e.g., genomics with linkage disequilibrium, spatially correlated environmental measurements, or multi-task learning with shared representations). We discuss the extension to dependent tests below:
>
> **Extension to dependent tests.** Our workflow — compute statistics, select a subset, report a calibrated lower bound — naturally accommodates correlated tests, but the methodological development changes substantially in both the optimization and the calibration steps.
>
> *Optimization.* Under independence, the denominator of the test statistic $T(S) = \sum_i \hat{\theta}_i z_i / \sqrt{\sum_i \hat{\sigma}_i^2 z_i}$ is additive in $z_i$, which enables the LP/MILP reformulation and the sign-screening property (negative-$\hat{\theta}_i$ items can be safely excluded). With correlated tests, the denominator becomes $\sqrt{z^\top \hat{C} z}$ where $\hat{C}$ is the joint covariance matrix — this is no longer additive in $z_i$, invalidating the LP reformulation. Moreover, sign screening no longer holds: an item with $\hat{\theta}_i < 0$ may improve the ratio $T(S)$ if it has sufficiently negative covariance with the selected items, reducing the denominator more than the numerator. The optimization thus becomes a fractional binary program with a quadratic denominator, requiring different algorithmic tools.
>
> *Bootstrap calibration.* The independent bootstrap $N(0, \text{diag}(\hat{\sigma}_i^2))$ is replaced by sampling from $N(0, \hat{C})$, and each bootstrap replicate requires solving the (harder) correlated optimization.
>
> *Finite-sample guarantees.* Our finite-sample coverage analysis (Theorem 3, Corollary 1) exploits independence across tests to decompose the bootstrap approximation error into per-test marginal non-Gaussianity terms. Under dependence, this decomposition breaks down — the error would additionally depend on the accuracy of the covariance estimate $\hat{C}$ and the dependence structure itself.
>
> In summary, extending to correlated tests is a meaningful direction for future work, but the methodological challenges are qualitatively different from the independent case we study.
>
> ---

---

> > ### Author Rebuttal · Reviewer_APZ1 · 2026-04-02
> >
> > Thank you for the rebuttal. The response helps clarify the scope of the paper and addresses part of my concern about the independence assumption, but I still have several follow-up questions.
> >
> > - **On the independence assumption:** the rebuttal explains why independence is natural in some application settings and gives useful examples. However, it does not provide either theoretical extensions or empirical robustness checks for dependent tests, which was one of my main concerns. I would still like more guidance on how practitioners should assess applicability when correlations among tests are present.
> >
> > - **On computational scalability:** my question about runtime and solver behavior is still largely unanswered. In particular, I was looking for concrete runtime measurements for the LP/MILP optimization across different values of \(m\) and bootstrap sizes \(B\), as well as some indication of whether the method remains practical when \(m\) is substantially larger than the scales studied in the paper.
> >
> > - **On experimental scale:** the rebuttal does not appear to add evidence beyond the relatively small-scale settings in the submission. I still find it difficult to judge how the method would behave in larger and more realistic ML problems, where the number of tests can be much higher.
> >
> > - **On the choice of bootstrap size:** my concern about the practical choice of \(B\) also remains. While the theory includes a finite-\(B\) term, I did not see a practical justification for the value used in experiments or a sensitivity analysis showing how performance changes with \(B\).
> >
> > - **On broader applicability:** the rebuttal gives helpful examples of domains where independence is plausible, but it does not yet clarify how broadly the proposed framework can be applied in modern ML settings where structured dependence is common.
> >
> > Overall, the rebuttal is helpful in clarifying the intended setting of the paper, but several of my original concerns remain only partially addressed, and I would welcome further clarification on the points above.

---

> > > ### Author Response · Authors · 2026-04-05
> > >
> > > ## Independence
> > >
> > > We provide an empirical robustness check. We run SubsetMaxT on correlated data, generating equicorrelated noise via a one-factor model ($\epsilon_j = \sqrt{\rho}Z + \sqrt{1-\rho}W_j$), giving all-pairs correlation $+\rho$. For mixed-sign correlations, we flip signs of half the components ($+\rho$ within halves, $-\rho$ between).
> > >
> > > | $\rho$ | Corr Type | SubsetMaxT Coverage | SubsetMaxT Bound | Scheffe Bound | Bonferroni Bound |
> > > |--------|-----------|---------------------|------------------|---------------|------------------|
> > > | 0 | none | 1.000 | 3.017 | 1.405 | 2.847 |
> > > | 0.1 | positive | 1.000 | 3.000 | 0.893 | 2.831 |
> > > | 0.2 | positive | 1.000 | 2.991 | 0.442 | 2.821 |
> > > | 0.5 | positive | 0.995 | 2.969 | −0.683 | 2.798 |
> > > | 0.8 | positive | 0.995 | 2.960 | −1.597 | 2.787 |
> > > | 0.5 | mixed | 0.995 | 2.965 | −0.673 | 2.794 |
> > > | 0.8 | mixed | 0.995 | 2.954 | −1.580 | 2.781 |
> > >
> > > ($m=100$, $k=5$.)
> > >
> > > SubsetMaxT maintains $\geq 99.5\\%$ coverage (nominal value $95\\%$) even at $\rho = 0.8$, demonstrating the robustness against the presence of correlation.
> > >
> > > ---
> > >
> > > ## Computational Scalability
> > >
> > > Runtime ($B=50$ with $10$ parallel workers) for our LP, MILP and new FPTAS approaches:
> > >
> > > | $m$ | LP | MILP ($k=5$) | FPTAS ($k=5$) |
> > > |-----|-----------|--------------|---------------|
> > > | 30 | 0.07s | 0.32s | 0.03s |
> > > | 100 | 0.45s | 15.9s | 0.04s |
> > > | 1,000 | 89s | — | 0.26s |
> > > | 5,000 | 10.8h | — | 1.18s |
> > >
> > > where — means the method is taking too long. It shows that LP is tractable for $m$ up to thousands but the MILP approach for constrained case quickly becomes computationally infeasible. We will add the following remedy for this:
> > >
> > > **FPTAS for the constrained case (MILP).** The t-statistic $T(S) = P/\sqrt{Q}$ depends on only two sufficient statistics $(P, Q) = (\sum_{i \in S} \hat\theta_i, \sum_{i \in S} \hat\sigma_i^2)$. This reduces the combinatorial optimization to a problem structurally analogous to a knapsack with a nonlinear ratio objective. For cardinality constraints ($|S| \leq k$), the cardinality serves as a DP dimension, so only one of $P,Q$ needs to be discretized — the other is tracked exactly as the DP value. We will show that standard knapsack FPTAS techniques (rounding + DP) then yield an $O(mk^3/\epsilon)$-time algorithm with $(1-\epsilon)$-approximation guarantee. By a similar idea, an FPTAS is also available for downward-closed families described by a single weight constraint ($a^T z \leq b$), where both $P$ and $Q$ are discretized and the DP tracks the minimum constraint weight — directly analogous to the classical 1-D knapsack FPTAS. The output is rescaled by $1/(1-\epsilon)$ to ensure the bootstrap critical value is not underestimated, preserving valid inference with controlled conservatism. In all experiments we use $\epsilon = 0.01$.
> > >
> > > **SubsetMaxT_FPTAS vs. baselines for large $m$**:
> > >
> > > | $m$ | $k$ | Method | Coverage | Bound Mean | Runtime (s) |
> > > |-----|-----|--------|----------|------------|-------------|
> > > | 500 | 5 | SubsetMaxT_FPTAS | 1.00 | 0.124 | 0.150 |
> > > | 500 | 5 | Scheffe | 1.00 | 0.116 | 0.001 |
> > > | 500 | 5 | Bonferroni | 1.00 | 0.051 | <0.001 |
> > > | 1,000 | 10 | SubsetMaxT_FPTAS | 1.00 | 0.323 | 0.670 |
> > > | 1,000 | 10 | Scheffe | 1.00 | 0.309 | 0.001 |
> > > | 1,000 | 10 | Bonferroni | 1.00 | 0.089 | <0.001 |
> > > | 5,000 | 10 | SubsetMaxT_FPTAS | 1.00 | 0.232 | 3.407 |
> > > | 5,000 | 10 | Scheffe | 1.00 | 0.222 | 0.005 |
> > > | 5,000 | 10 | Bonferroni | 1.00 | 0.014 | <0.001 |
> > >
> > > SubsetMaxT_FPTAS consistently produces the tightest bounds. We will add these results to the revision. This result also demonstrates the performance of our method when the scale of tests ($m$) is in thousands. Having said that, we have to acknowledge that even with the LP or FPTAS solving the optimization at even larger scale is challenging. However, given that this optimization is already nontrivial and challenging for moderate $m$, the case of large scale of $m$ is worth a future investigation.
> > >
> > > ---
> > >
> > > ## Justification of Bootstrap Size
> > >
> > > In the ideal case where test statistics are exactly Gaussian with known variances, the maximum t-statistic $\max_{S \in \mathcal{A}} T(S; \hat{\theta}-\theta, \hat{\Sigma})$ and the $B$ bootstrap samples are i.i.d.. By exchangeability, using the $\lceil(1-\alpha)(B+1)\rceil$-th order statistic as the critical value gives exact coverage $\lceil(1-\alpha)(B+1)\rceil/(B+1) \geq 1-\alpha$, valid as soon as $B \geq (1-\alpha)/\alpha = 19$. Small $B$ does not invalidate coverage — it only causes overcoverage (wider bounds): the gap is $\leq 1/(B+1)$, shrinking with $B$. Although this is true only in the ideal case, it provides insight into the behavior in the realistic case, as seen in the following result.
> > >
> > > | $B$ | Coverage | Bound Mean (LP, $k=m=100$) | Bound Mean (FPTAS, $k=5$) |
> > > |-----|----------|----------------------------|---------------------------|
> > > | 30 | 1.00 | 5.182 | 2.990 |
> > > | 50 | 1.00 | 5.243 | 3.018 |
> > > | 200 | 1.00 | 5.298 | 3.053 |
> > > | 1,000 | 1.00 | 5.303 | 3.055 |
> > >
> > > Bound mean stabilizes by $B = 50$.

---

### Official Review · Reviewer_FeoM · 2026-03-12

**Soundness:** 3
**Presentation:** 3
**Significance:** 3
**Originality:** 3
**Overall Recommendation:** 5
**Confidence:** 4

**Summary:**

This paper considers the construction of simultaneous confidence bounds for aggregated effects across selected subsets. This addresses the critical challenge of post-selection bias in high-dimensional screening. The authors propose a bootstrap-calibrated procedure for the null distribution and SubsetMaxT, which targets the maximum normalized aggregated effect over a downward-closed family of subsets to provide valid post hoc inference. To overcome the combinatorial complexity of the required maximization, the problem is elegantly reduced to a weighted densest-subgraph formulation. This allows for exact optimization via Linear Programming (LP) for unconstrained families or Mixed-Integer Linear Programming (MILP) for those with additional linear constraints. Both theoretical analysis and empirical evaluations demonstrate that this approach yields significantly tighter confidence bounds than classical methods like Bonferroni or Scheffé adjustments, while maintaining rigorous coverage guarantees for complex, non-Gaussian data structures.

**Compliance With Llm Reviewing Policy:**

Affirmed.

**Final Justification:**

The authors have adequately answered my question. I am happy to keep my score.

**Key Questions For Authors:**

* Can the authors discuss specific examples with background in literature and comparisons against exisiting methods?

**Limitations:**

yes

**Strengths And Weaknesses:**

Strength
* Addresses both theoretical and computational challenges for simultaneous confidence bounds for aggregated effects across selected subsets
* The framework is elegant and general

Weakness
* Lack of understanding between the interplay of the size of the study for each effect and the number of studies
* Theoretical comparisons against other numerical competitors in specific examples of the general framework

---

> ### Author Rebuttal · Authors · 2026-03-31
>
> We thank the reviewer for the encouraging assessment.
>
> ## Interplay of Study Size and Number of Studies
>
> The Corollary to Theorem 3 makes the interplay explicit. The coverage error rate is $O\big((m + \log|\mathcal{A}|) / \sqrt{n_{\min}}\big)$, where $m$ is the number of tests and $n_{\min} = \min_i n_i$ is the smallest per-test sample size. This reveals a clear trade-off:
>
> - **Per-test sample size $n_i$**: larger $n_i$ improves the Gaussian approximation of each $\hat{\theta}_i$ and the accuracy of variance estimates $\hat{\sigma}_i^2$. The error decays as $1/\sqrt{n_{\min}}$.
> - **Number of tests $m$**: more tests increase the complexity of the subset family ($|\mathcal{A}|$ can grow exponentially in $m$) and the marginal non-Gaussianity term (which sums over $m$ tests). The error grows as $m / \sqrt{n_{\min}}$.
>
> The method is most effective when $m = o(\sqrt{n_{\min}})$ — i.e., moderate number of tests with large per-test samples. We note that the condition $m = o(\sqrt{n_{\min}})$ is a sufficient condition for the theoretical guarantee; in practice, our method works well even when $m$ is larger than $n$, maintaining valid coverage empirically.
>
> We will add a paragraph to the paper explicitly discussing this trade-off.

---

> > ### Author Rebuttal · Reviewer_FeoM · 2026-04-03
> >
> > The authors have adequately answered all my comments, and I have no further questions.

---

### Official Review · Reviewer_ExxP · 2026-03-12

**Soundness:** 3
**Presentation:** 3
**Significance:** 3
**Originality:** 3
**Overall Recommendation:** 4
**Confidence:** 5

**Summary:**

This submission addresses the construction of valid simultaneous confidence bounds for the sum of effects over a data-dependent subset. The proposed method, SubsetMaxT, employs calibration via a multiplier bootstrap, which enhances its reliability. Its key contribution is an exact and efficient algorithm for the required combinatorial optimization (maximizing the studentized sum). This is achieved by linking the problem to a densest-subgraph formulation and solving it via a linear program or a mixed-integer linear program for constrained families. The method is supported by theoretical coverage guarantees and empirical demonstrations of its tightness relative to the Scheffe and Bonferroni baselines.

**Compliance With Llm Reviewing Policy:**

Affirmed.

**Key Questions For Authors:**

For large mm and complex constraints, solving a MILP for each bootstrap sample could be a bottleneck. Could the authors comment on the practical scalability of this approach?

How might the choice of subset search algorithm (e.g., one that selects larger subsets) affect the relative performance of SubsetMaxT compared to the baselines?

All tables show coverage 1.000 or very close, nearly perfect. The paper stated SubsetMaxT delivers consistently tighter bounds than Scheffe and Bonferroni. Where is this claim read from in Table 8.

**Limitations:**

The method assumes independent tests (diagonal covariance), which is a strong limitation for applications with correlated data, like genomics.

The method requires the family of subsets to be downward-closed, which excludes some potentially interesting but non-monotone families.

The finite-sample bound is qualitative, involving unknown constants, and suggests the method is most effective when the number of tests mm is small relative to per-test sample sizes.

**Strengths And Weaknesses:**

Soundness: The theoretical results are rigorous, with clear proofs for optimization equivalences and a detailed finite-sample bound for the bootstrap's coverage error.
Presentation: The paper is well-organized, logically flows from problem to solution, and includes a thorough experimental section validating its practical value.
Significance: By focusing on aggregated subset sums, the method yields much tighter bounds than generic post-selection approaches, and its efficient algorithm makes it practically applicable.
Originality: The novel connection between post-selection inference and the densest-subgraph problem is elegant and provides a powerful new algorithmic lens.

Weaknesses: The leap from the subset-sum objective to the linear program in Theorem 4.3 is dense. The main text would benefit from a high-level intuition for this transformation to improve accessibility. The paper introduces a MILP for general constraints but does not discuss its scalability. For large problems, solving a MILP for each bootstrap sample could be a major practical limitation. The experiments use a single greedy search algorithm. The paper does not explore how different subset selection methods might interact with the performance of the proposed bounds.

---

> ### Author Rebuttal · Authors · 2026-03-31
>
> We thank the reviewer for the positive evaluation and constructive suggestions. We address each point below.
>
> ## LP Transformation Intuition
>
> We agree the LP reformulation deserves more intuition. The derivation has two steps:
>
> **Step 1: Variable substitution.** For any $z_i \in \{0,1\}$, define $u_i = z_i / \sum_j \hat\sigma_j^2 z_j$ and $v_{ij} = \min(u_i, u_j)$ (which equals $z_i z_j / \sum_j \hat\sigma_j^2 z_j$ since $z_i \in \{0,1\}$). Then $\sum \hat\sigma_i^2 u_i = 1$ and the objective $\sum_{i,j} \hat\theta_i \hat\theta_j v_{ij} = \sum_{i,j} \hat\theta_i \hat\theta_j z_i z_j / \sum_j \hat\sigma_j^2 z_j$. So every binary $z$ maps to a feasible LP solution with the correct objective value.
>
> **Step 2: The relaxation does not inflate the optimum.** The LP allows $u_i$ to take arbitrary nonneg values (not just $\{0, 1/ \sum_j \hat\sigma_j^2 z_j\}$), so it could in principle find a fractional solution with higher objective. But it cannot: for any LP optimum $u^*$, thresholding at any level $r\geq 0$ produces a valid subset $S(r) = \{i : u_i \ge r\}$, and a parametric averaging argument (analogous to Charikar 2000 for the classic densest subgraph LP) shows that the best threshold achieves an objective at least as large as the LP value. So the LP optimum equals the discrete optimum.
>
> We will add this two-step intuition to the main text.
>
> ## Table 8 Comparison of Bound Tightness
>
> The tightness comparison in Table 8 is read from the "Bound Mean" columns. Since we are constructing confidence *lower* bounds, a larger bound mean is better (tighter) as long as coverage remains $\ge 95\\%$. At matched coverage levels, SubsetMaxT produces the largest (tightest) bound mean in four out of six settings. In the remaining two cases ($k = 1$), Bonferroni is slightly better — this is expected because $k = 1$ restricts the family to singletons, where the subset-maximum statistic reduces to the individual maximum and Bonferroni (which allocates $\alpha / m$ per test) is near-optimal for independent tests. We will clarify this in the text.

---

> > ### Author Rebuttal · Reviewer_ExxP · 2026-04-03
> >
> > It looks the authors missed some points I raised.

---

> > > ### Author Response · Authors · 2026-04-05
> > >
> > > ## Computational Scalability
> > >
> > > See our second response to Reviewer APZ1 for runtime analysis and the new FPTAS algorithm (named SubsetMaxT_FPTAS), as well as additional experimental results for the FPTAS with much larger $m$. Here we also report SubsetMaxT_FPTAS ($\epsilon = 0.01$) results for $m = 30$–$100$ under sparse signal with homogeneous lognormal noise and $B=50$, the same setting as Table 1 in the paper, so the bounds can be directly compared with those there:
> > >
> > > | $m$ | $k$ | Coverage | Bound Mean | Runtime (s) |
> > > |-----|-----|----------|------------|-------------|
> > > | 30 | 5 | 1.00 | 1.361 | 0.033 |
> > > | 30 | 10 | 1.00 | 1.288 | 0.044 |
> > > | 50 | 5 | 1.00 | 2.232 | 0.036 |
> > > | 50 | 10 | 1.00 | 2.070 | 0.056 |
> > > | 100 | 5 | 1.00 | 4.301 | 0.043 |
> > > | 100 | 10 | 1.00 | 7.042 | 0.079 |
> > >
> > > Coverage is 100% with runtime under 0.08s for all settings. Comparing with Table 1, SubsetMaxT_FPTAS produces bounds close to the exact method SubsetMaxT while being significantly faster. Notably, at $m = 100$, SubsetMaxT_FPTAS now outperforms both Scheffe and Bonferroni baselines in bound tightness. We will add these results to the revision.
> > >
> > > ---
> > >
> > > ## Experiments with Different Subset Selection Algorithms
> > >
> > > We now include experiments with multiple subset selection algorithms: **TstatOrderSearch** (sort by $t_i = \hat{\theta}_i/\hat{\sigma}_i$, scan prefixes), **ForwardStepwiseSearch** (greedy forward selection maximizing $T(S)$), and **GLRScanSearch** (original). Coverage holds *regardless* of search algorithm, since the bootstrap critical value protects against the worst-case subset in $\mathcal{A}$. Here are the results:
> > >
> > > **TstatOrderSearch:**
> > >
> > > | $m$ | $k$ | Method | Coverage | Bound Mean |
> > > |-----|-----|--------|----------|------------|
> > > | 30 | 5 | SubsetMaxT | 1.00 | 2.105 |
> > > | 30 | 5 | SubsetMaxT_FPTAS | 1.00 | 2.100 |
> > > | 30 | 5 | Scheffe | 1.00 | 2.059 |
> > > | 30 | 5 | Bonferroni | 1.00 | 2.019 |
> > > | 50 | 10 | SubsetMaxT | 1.00 | 1.763 |
> > > | 50 | 10 | SubsetMaxT_FPTAS | 1.00 | 1.748 |
> > > | 50 | 10 | Scheffe | 1.00 | 1.583 |
> > > | 50 | 10 | Bonferroni | 1.00 | 1.301 |
> > > | 100 | 10 | SubsetMaxT_FPTAS | 1.00 | 5.924 |
> > > | 100 | 10 | Scheffe | 1.00 | 5.694 |
> > > | 100 | 10 | Bonferroni | 1.00 | 5.232 |
> > > | 500 | 10 | SubsetMaxT_FPTAS | 1.00 | 9.438 |
> > > | 500 | 10 | Scheffe | 1.00 | 9.309 |
> > > | 500 | 10 | Bonferroni | 1.00 | 9.053 |
> > > | 1000 | 10 | SubsetMaxT_FPTAS | 1.00 | 11.224 |
> > > | 1000 | 10 | Scheffe | 1.00 | 11.123 |
> > > | 1000 | 10 | Bonferroni | 1.00 | 10.920 |
> > > | 5000 | 10 | SubsetMaxT_FPTAS | 1.00 | 11.130 |
> > > | 5000 | 10 | Scheffe | 1.00 | 11.074 |
> > > | 5000 | 10 | Bonferroni | 1.00 | 10.965 |
> > >
> > > **ForwardStepwiseSearch:**
> > >
> > > | $m$ | $k$ | Method | Coverage | Bound Mean |
> > > |-----|-----|--------|----------|------------|
> > > | 30 | 5 | SubsetMaxT | 1.00 | 1.891 |
> > > | 30 | 5 | SubsetMaxT_FPTAS | 1.00 | 1.887 |
> > > | 30 | 5 | Scheffe | 1.00 | 1.851 |
> > > | 30 | 5 | Bonferroni | 1.00 | 1.816 |
> > > | 50 | 10 | SubsetMaxT | 1.00 | 1.379 |
> > > | 50 | 10 | SubsetMaxT_FPTAS | 1.00 | 1.369 |
> > > | 50 | 10 | Scheffe | 1.00 | 1.254 |
> > > | 50 | 10 | Bonferroni | 1.00 | 1.056 |
> > > | 100 | 10 | SubsetMaxT_FPTAS | 1.00 | 4.629 |
> > > | 100 | 10 | Scheffe | 1.00 | 4.464 |
> > > | 100 | 10 | Bonferroni | 1.00 | 4.134 |
> > > | 500 | 10 | SubsetMaxT_FPTAS | 1.00 | 6.936 |
> > > | 500 | 10 | Scheffe | 1.00 | 6.842 |
> > > | 500 | 10 | Bonferroni | 1.00 | 6.658 |
> > > | 1000 | 10 | SubsetMaxT_FPTAS | 1.00 | 8.707 |
> > > | 1000 | 10 | Scheffe | 1.00 | 8.629 |
> > > | 1000 | 10 | Bonferroni | 1.00 | 8.472 |
> > > | 5000 | 10 | SubsetMaxT_FPTAS | 1.00 | 10.031 |
> > > | 5000 | 10 | Scheffe | 1.00 | 9.981 |
> > > | 5000 | 10 | Bonferroni | 1.00 | 9.883 |
> > >
> > > Coverage is 100% in all cases and for both search algorithms. SubsetMaxT and SubsetMaxT_FPTAS ($\epsilon = 0.01$) consistently produce the tightest bounds, outperforming both Scheffe and Bonferroni across all settings. We will add these results to the revision.
> > >
> > > ---
> > >
> > > ## Independence Assumption
> > >
> > > See our response to Reviewer APZ1.
> > >
> > > ---
> > >
> > > ## Qualitative Finite-Sample Bound
> > >
> > > The bound in Theorem 3 involves implicit constants but reveals the key structural dependencies: the coverage error decomposes into (i) marginal non-Gaussianity, (ii) variance estimation error, and (iii) bootstrap discretization. The Corollary yields the rate $\mathcal{E} = O((m + \log|\mathcal{A}|)/\sqrt{n_{\min}})$, clarifying that the method is effective when $m$ is small relative to $\sqrt{n_{\min}}$ — natural in our experimented applications. Empirically, however, the method works well even when $m \gg n$: in the tables above, coverage remains 100% at $m = 5{,}000$ with sample size $n = 50$ for each test, suggesting the finite-sample bound is conservative and the method may be effective at much larger $m$.

---

### Official Review · Reviewer_G2q3 · 2026-03-13

**Soundness:** 3
**Presentation:** 3
**Significance:** 3
**Originality:** 3
**Overall Recommendation:** 4
**Confidence:** 3

**Summary:**

The paper develops a new method for aggregated effects over complex subset families, with provable and efficient confidence bounds. Namely, it proposes to calibrate the maximum normalized/studentized aggregated effect over any downward-closed subset family --- and in view of the combinatorially hard nature of this maximization problem, develops a novel method that still allows for scalably solving it. In particular, after certain transformations, the problem is shown to be possible to rewrite as a MILP, therefore enabling computationally efficient evaluation. Furthermore, novel finite-sample coverage guarantees are developed for the setup. Finally, extensive empirical evaluation of the method is provided over a variety of datasets with different sparsity, homogeneity, and other properties.

**Compliance With Llm Reviewing Policy:**

Affirmed.

**Final Justification:**

I confirm my positive opinion of this paper (which the rebuttal did not change). I have also carefully familiarized myself with the rest of the author-reviewer discussions, and believe that while several further points were raised, such as relaxing (independence etc.) assumptions underlying the method, testing its computational tractability more rigorously, etc., but ultimately the authors' rebuttals added further clarity and detail and underscored that the paper is a valid and worthy contribution.

My personal opinion rests on the technical non-triviality/ingenuity of the paper's proofs via graphs; and I believe it is on balance worth supporting despite some stylized aspects to its contribution.

**Key Questions For Authors:**

N/A

**Limitations:**

yes

**Strengths And Weaknesses:**

Overall, I believe that this paper is a worthy contribution to the literature.

At its heart, the core contribution is based on a simple combinatorial idea --- yet noticing this idea is nontrivial, and various technical details of the analysis require care and precision. The idea is that after transforming/rewriting a normalized version of the maximum summed effect over the subset family, it reveals itself as a (constrained) weighted densest subgraph problem, which is a known combinatorial setting in the algorithmic literature, yielding insight that enables scalable evaluation.

While one could argue that this reformulation is a simple observation conditional on some familiarity with graph algorithmics, I would argue that the ensemble of identifying the overall setting, singling out the normalized aggregated statistics' monotonicity/sign-positivity properties, and then noticing the physical meaning behind the variables, is quite an interesting and non-trivial contribution nonetheless.

Moreover, finite-sample coverage guarantees for the resulting calibration procedure are not completely obvious and require a hairy custom derivation: as the authors correctly point out, standard Gaussian approximation routes are not directly useful because the subset family can be exponentially large, and therefore its structure must be utilized instead.

Finally, empirical evaluation is reasonably comprehensive and covers explorations of heterogeneity/sparsity; A/B testing datasets such as Criteo; and ML on many distinct tabular datasets --- and I found these illustrations to be adequately motivated with respect to the paper's studied setup with aggregation over downward closed subset systems. Indeed, the proposed procedure (SubsetMaxT) is found to be better than the Scheffe and Bonferroni alternatives --- this is not surprising since it directly targets the subset structure, but reassuring nonetheless.

In all, I believe this paper provides a conceptually simple but meaningful contribution to the literature, and therefore support it in its current form.

---

> ### Author Rebuttal · Authors · 2026-03-31
>
> We sincerely thank the reviewer for the positive and supportive assessment. We are glad the reviewer found our combinatorial reformulation, finite-sample coverage analysis, and empirical evaluation to be meaningful contributions. We appreciate the observation that the ensemble of identifying the setting, exploiting sign-positivity, and connecting to the densest subgraph literature is non-trivial.

---

> > ### Author Rebuttal · Reviewer_G2q3 · 2026-04-04
> >
> > Thanks again for your rebuttals to all the reviewers' feedback. I carefully read all of them, and despite learning about some points of criticism put forth by other reviewers (some of which basically pertain to whether/how easily the method can be generalized beyond its current simple setting), on balance I continue to stand by my positive evaluation of the paper, due to its interesting/nontrivial mathematical approach to the problem at hand.

---

### Decision · Program_Chairs · 2026-04-30

**Decision:**

Accept (regular)

**Comment:**

This submission is recommended for Acceptance as all reviewers praised its elegant and novel reformulation of post-selection inference as a Weighted Densest Subgraph problem. The authors' rebuttal successfully addressed core concerns regarding computational scalability. Ultimately, the reviewers agree that the paper's technical ingenuity and improved bound tightness represent a significant and well-supported contribution.